# GRAPHCARE: ENHANCING HEALTHCARE PREDICTIONS WITH PERSONALIZED KNOWLEDGE GRAPHS

**Pengcheng Jiang**[*]  **Cao Xiao**[†]  **Adam Cross**[‡]  **Jimeng Sun**[*]
[*]University of Illinois Urbana-Champaign  [†]GE HealthCare  [‡]OSF HealthCare

## ABSTRACT

Clinical predictive models often rely on patients' electronic health records (EHR), but integrating medical knowledge to enhance predictions and decision-making is challenging. This is because personalized predictions require personalized knowledge graphs (KGs), which are difficult to generate from patient EHR data. To address this, we propose GRAPHCARE, a framework that uses external KGs to improve EHR-based predictions. Our method extracts knowledge from large language models (LLMs) and external biomedical KGs to build patient-specific KGs, which are then used to train our proposed Bi-attention AugmenTed (BAT) graph neural network (GNN) for healthcare predictions. On two public datasets, MIMIC-III and MIMIC-IV, GRAPHCARE surpasses baselines in four vital healthcare prediction tasks: mortality, readmission, length of stay (LOS), and drug recommendation. On MIMIC-III, it boosts AUROC by 17.6% and 6.6% for mortality and readmission, and F1-score by 7.9% and 10.8% for LOS and drug recommendation, respectively. Notably, GRAPHCARE demonstrates a substantial edge in scenarios with limited data. Our findings highlight the potential of using external KGs in healthcare prediction tasks and demonstrate the promise of GRAPHCARE in generating personalized KGs for promoting personalized medicine.

## 1 INTRODUCTION

The digitization of healthcare systems has led to the accumulation of vast amounts of electronic health record (EHR) data that encode valuable information about patients, treatments, etc. Machine learning models have been developed on these data and demonstrated huge potential for enhancing patient care and resource allocation via predictive tasks, including mortality prediction (Blom et al., 2019; Courtright et al., 2019), length-of-stay (LOS) estimation (Cai et al., 2015; Levin et al., 2021), readmission prediction (Ashfaq et al., 2019; Xiao et al., 2018), and drug recommendations (Bhoi et al., 2021; Shang et al., 2019b).

To improve predictive performance and integrate expert knowledge with data insights, clinical knowledge graphs (KGs) were adopted to complement EHR modeling (Chen et al., 2019; Choi et al., 2020; Rotmensch et al., 2017). These KGs represent medical concepts (e.g., diagnoses, procedures, drugs) and their relationships, enabling effective learning of patterns and dependencies. However, existing approaches mainly focus on simple hierarchical relations (Choi et al., 2017; 2018; 2020) rather than leveraging comprehensive relationships among biomedical entities despite incorporating valuable contextual information from established biomedical knowledge bases (e.g., UMLS (Bodenreider, 2004)) could enhance predictions. Moreover, large language models (LLMs) such as GPT (Brown et al., 2020; Chowdhery et al., 2022; Luo et al., 2022; OpenAI, 2023) pre-trained on web-scale biomedical literature could serve as alternative resources for extracting clinical knowledge given their remarkable reasoning abilities on open-world data. There is a substantial body of existing research demonstrating their potential use as knowledge bases (Lv et al., 2022; Petroni et al., 2019; AlKhamissi et al., 2022).

To fill the gap in personalized medical KGs, we propose to leverage the exceptional reasoning abilities of LLMs to extract and integrate personalized KG from open-world data. Our proposed method GRAPHCARE (Personalized **Graph**-based Health**Care** Prediction) is a framework designed to generate patient-specific KGs by effectively harnessing the wealth of clinical knowledge. As shown

---

[*]{pj20, jimeng}@illinois.edu

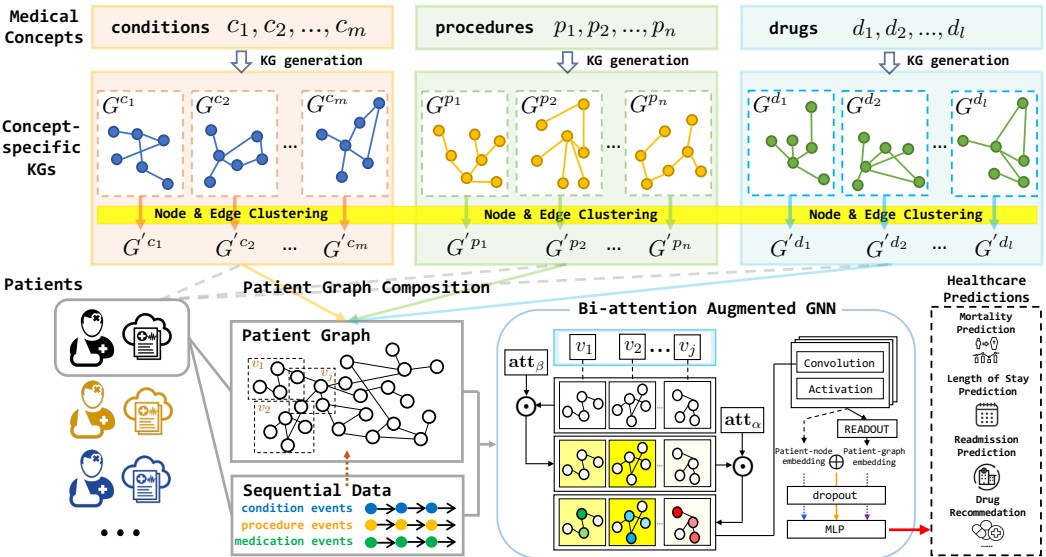

Figure 1: **Overview of GRAPHCARE**. *Above*: Given a patient record consisting of conditions, procedures and medications, we generate a concept-specific KG for each medical concept, by knowledge probing from a LLM and subgraph sampling from an existing KG; and we perform node and edge clustering among all graphs (§3.1). *Below*: For each patient, we compose a patient-specific graph by combining the concept-specific KGs associated with them and make the graph temporal with sequential data across patient's visits (§3.2). To utilize the patient graph for predictions, we employ a bi-attention augmented graph neural network (GNN) model, which highlights essential visits and nodes with attention weights (§3.3). With three types of patient representations (patient-node, patient-graph, and joint embeddings), GRAPHCARE is capable of handling a variety of healthcare predictions (§3.4).

in Figure 1, our patient KG generation module first takes medical concepts as input and generates concept-specific KGs by prompting LLMs or retrieving subgraphs from existing graphs. It then clusters nodes and edges to create a more aggregated KG for each medical concept. Next, it constructs a personalized KG for each patient by merging their associated concept-specific KGs and incorporating temporal information from their sequential visit data. These patient-specific graphs are then fed into our **B**i-attention **A**ugmen**T**ed (BAT) graph neural network (GNN) for diverse downstream prediction tasks.

We evaluated the effectiveness of GRAPHCARE using two widely-used EHR datasets, MIMIC-III (Johnson et al., 2016) and MIMIC-IV (Johnson et al., 2020). Through extensive experimentation, we found that GRAPHCARE outperforms several baselines, while BAT outperforms state-of-the-art GNN models (Veličković et al., 2017; Hu et al., 2019; Rampášek et al., 2022) on four common healthcare prediction tasks: mortality prediction, readmission prediction, LOS prediction, and drug recommendation. Our experimental results demonstrate that GRAPHCARE, equipped with the BAT, achieves average AUROC improvements of 17.6%, 6.6%, 4.1%, 2.1% and 7.9%, 3.8%, 3.5%, 1.8% over all baselines on MIMIC-III and MIMIC-IV, respectively. Furthermore, our approach requires significantly fewer patient records to achieve comparable results, providing compelling evidence for the benefits of integrating open-world knowledge into healthcare predictions.

## 2   RELATED WORKS

**Clinical Predictive Models.** EHR data has become increasingly recognized as a valuable resource in the medical domain, with numerous predictive tasks utilizing this data (Ashfaq et al., 2019; Bhoi et al., 2021; Blom et al., 2019; Cai et al., 2015). A multitude of deep learning models have been designed to cater to this specific type of data, leveraging its rich, structured nature to achieve enhanced performance (Shickel et al., 2017; Miotto et al., 2016; Choi et al., 2016c;a;b; Shang et al., 2019b; Yang et al., 2021a; Choi et al., 2020; Zhang et al., 2020; Ma et al., 2020b;a; Gao et al., 2020; Yang et al., 2023b). Among these models, some employ a graph structure to improve prediction accuracy, effectively capturing underlying relationships among medical entities (Choi et al., 2020; Su et al., 2020; Zhu & Razavian, 2021; Li et al., 2020; Xie et al., 2019; Lu et al., 2021a; Yang et al.,

2023b; Shang et al., 2019b). However, a limitation of these existing works is that they do not link the local graph to an external knowledge base, which contains a large amount of valuable relational information (Lau-Min et al., 2021; Pan & Cimino, 2014). We propose to create a customized knowledge graph for each medical concept in an open-world setting by probing relational knowledge from either LLMs or KGs, enhancing its predictive capabilities for healthcare.

**Personalized Knowledge Graphs.** Personalized KGs have emerged as promising tools for improving healthcare prediction (Ping et al., 2017; Gyrard et al., 2018; Shirai et al., 2021; Rastogi & Zaki, 2020; Li et al., 2022). Previous approaches such as GRAM (Choi et al., 2017) and its successors (Ma et al., 2018; Shang et al., 2019a; Yin et al., 2019; Panigutti et al., 2020; Lu et al., 2021b) incorporated hierarchical graphs to improve predictions of deep learning-based models; however, they primarily focus on simple parent-child relationships, overlooking the rich complexities found in large knowledge bases. MedML (Gao et al., 2022) employs graph data for COVID-19 related prediction. However, the KG in this work has a limited scope and relies heavily on curated features. To bridge these gaps, we introduce two methods for creating detailed, personalized KGs using open sources. The first solution is prompting (Liu et al., 2023) LLMs to generate KGs tailored to medical concepts. This approach is inspired by previous research (Yao et al., 2019; Wang et al., 2020a; Chen et al., 2022; Lovelace & Rose, 2022; Chen et al., 2023; Jiang et al., 2023), showing that pre-trained LMs can function as comprehensive knowledge bases. The second method involves subgraph sampling from established KGs (Bodenreider, 2004), enhancing the diversity of the knowledge base.

**Attention-augmented GNNs.** Attention mechanisms (Bahdanau et al., 2014) have been widely utilized in GNNs to capture the most relevant information from the graph structure for various tasks (Veličković et al., 2017; Lee et al., 2018; Zhang et al., 2018; Wang et al., 2020b; Zhang et al., 2021a; Knyazev et al., 2019). The incorporation of attention mechanisms in GNNs allows for enhanced graph representation learning, which is particularly useful in the context of EHR data analysis (Choi et al., 2020; Lu et al., 2021b). In GRAPHCARE, we introduce a new GNN BAT leveraging both visit-level and node-level attention, edge weights, and attention initialization for EHR-based predictions with personalized KGs.

## 3 Personalized Graph-based HealthCare Prediction

In this section, we present GRAPHCARE, a comprehensive framework designed to generate personalized KGs and utilize them for healthcare predictions. It operates through three general steps:

*Step 1:* Generate concept-specific KGs for every medical concept using LLM prompts and by subsampling from existing KGs. Perform clustering on nodes and edges across these KGs.

*Step 2:* For each patient, merge relevant concept-specific KGs to form a personalized KG.

*Step 3:* Employ the novel Bi-attention Augmented (BAT) Graph Neural Network (GNN) to predict based on the personalized KGs.

### 3.1 Step 1: Concept-Specific Knowledge Graph Generation.

Denote a medical concept as $e \in \{\mathbf{c}, \mathbf{p}, \mathbf{d}\}$, where $\mathbf{c} = (c_1, c_2, ..., c_{|\mathbf{c}|})$, $\mathbf{p} = (p_1, p_2, ..., p_{|\mathbf{p}|})$, and $\mathbf{d} = (d_1, d_2, ..., d_{|\mathbf{d}|})$ correspond to sets of medical concepts for conditions, procedures, and drugs, with sizes of $|\mathbf{c}|$, $|\mathbf{p}|$, and $|\mathbf{d}|$, respectively. The goal of this step is to generate a KG $G^e = (\mathcal{V}^e, \mathcal{E}^e)$ for each medical concept $e$, where $\mathcal{V}^e$ represents nodes, and $\mathcal{E}^e$ denotes edges in the graph.

Our approach comprises two strategies: **(1) LLM-based KG extraction via prompting:** Utilizing a template with instruction, example, and prompt. For example, with an instruction "*Given a prompt, extrapolate as many relationships as possible of it and provide a list of updates*", an example "*prompt: systemic lupus erythematosus. updates: [systemic lupus erythematosus is, treated with, steroids]...*" and a prompt "*prompt: tuberculosis. updates:*", the LLM would respond with a list of KG triples such as "*[tuberculosis, may be treated with, antibiotics], [tuberculosis, affects, lungs]...*" where each triple contains a head entity, a relation, and a tail entity. Our curated prompts are detailed in Appendix D.1. After running $\chi$ times, we aggregate[1] and parse the outputs to form a KG for each medical concept, $G^e_{\text{LLM}(\chi)} = (\mathcal{V}^e_{\text{LLM}(\chi)}, \mathcal{E}^e_{\text{LLM}(\chi)})$. **(2) Subgraph sampling from existing**

---

[1]To address ethical concerns with LLM use, we collaborate with medical professionals to evaluate the extracted KG triples, which minimizes the risk of including any inaccurate or potentially misleading information.

**KGs:** Leveraging pre-existing biomedical KGs (Belleau et al., 2008; Bodenreider, 2004; Donnelly et al., 2006), we extract specific graphs for a concept via subgraph sampling. This involves choosing relevant nodes and edges from the primary KG. For this method, we first pinpoint the entity in the biomedical KG corresponding to the concept $e$. We then sample a $\kappa$-hop subgraph originated from the entity, resulting in $G^e_{\text{sub}(\kappa)} = (\mathcal{V}^e_{\text{sub}(\kappa)}, \mathcal{E}^e_{\text{sub}(\kappa)})$. We detail the sampling process in Appendix D.2. Consequently, for each medical concept, the KG is represented as $G^e = G^e_{\text{LLM}(\chi)} \cup G^e_{\text{sub}(\kappa)}$.

**Node and Edge Clustering.** Next, we perform clustering of nodes and edges based on their similarity, to refine the concept-specific KGs. The similarity is computed using the cosine similarity between their word embeddings. We apply the agglomerative clustering algorithm (Müllner, 2011) on the cosine similarity with a distance threshold $\delta$, to group similar nodes and edges in the global graph $G = (G^{e_1}, G^{e_2}, ..., G^{e(|\mathbf{c}|+|\mathbf{p}|+|\mathbf{d}|)})$ of all concepts. After the clustering process, we obtain $\mathcal{C}_\mathcal{V} : \mathcal{V} \to \mathcal{V}'$ and $\mathcal{C}_\mathcal{E} : \mathcal{E} \to \mathcal{E}'$ which map the nodes $\mathcal{V}$ and edges $\mathcal{E}$ in the original graph $G$ to new nodes $\mathcal{V}'$ and edges $\mathcal{E}'$, respectively. With these two mappings, we obtain a new global graph $G' = (\mathcal{V}', \mathcal{E}')$, and we create a new graph $G'^e = (\mathcal{V}'^e, \mathcal{E}'^e) \subset G'$ for each concept. The node embedding $\mathbf{H}^\mathcal{V} \in \mathbb{R}^{|\mathcal{V}'| \times w}$ and the edge embedding $\mathbf{H}^\mathcal{R} \in \mathbb{R}^{|\mathcal{E}'| \times w}$ are initialized by the averaged word embedding in each cluster, where $w$ denotes the dimension of the word embedding.

## 3.2 STEP 2: PERSONALIZED KNOWLEDGE GRAPH COMPOSITION

For each patient, we compose their personalized KG by merging the clustered KGs of their medical concepts. We create a patient node ($\mathcal{P}$) and connect it to its direct EHR nodes in the graph. The personalized KG for a patient can be represented as $G_{\text{pat}} = (\mathcal{V}_{\text{pat}}, \mathcal{E}_{\text{pat}})$, where $\mathcal{V}_{\text{pat}} = \mathcal{P} \cup \{\mathcal{V}'^{e_1}, \mathcal{V}'^{e_2}, ..., \mathcal{V}'^{e_\omega}\}$ and $\mathcal{E}_{\text{pat}} = \epsilon \cup \{\mathcal{E}'^{e_1}, \mathcal{E}'^{e_2}, ..., \mathcal{E}'^{e_\omega}\}$, with $\{e_1, e_2, ..., e_\omega\}$ being the medical concepts directly associated with the patient, $\omega$ being the number of concepts, and $\epsilon$ being the edge connecting $\mathcal{P}$ and $\{e_1, e_2, ..., e_\omega\}$. Further, as a patient is represented as a sequence of $J$ visits (Choi et al., 2016a), the *visit-subgraphs* for patient $i$ can be represented as $\{G_{i,1}, G_{i,2}, ..., G_{i,J}\} = \{(\mathcal{V}_{i,1}, \mathcal{E}_{i,1}), (\mathcal{V}_{i,2}, \mathcal{E}_{i,2}), ..., (\mathcal{V}_{i,J}, \mathcal{E}_{i,J})\}$ for visits $\{x_1, x_2, ..., x_J\}$ where $\mathcal{V}_{i,j} \subseteq \mathcal{V}_{\text{pat}(i)}$ and $\mathcal{E}_{i,j} \subseteq \mathcal{E}_{\text{pat}(i)}$ for $1 \leq j \leq J$. We introduce $\mathcal{E}_{\text{inter}}$ for the interconnectedness across these visit-subgraphs, defined as: $\mathcal{E}_{\text{inter}} = \{(v_{i,j,k} \leftrightarrow v_{i,j',k'}) | v_{i,j,k} \in \mathcal{V}_{i,j}, v_{i,j',k'} \in \mathcal{V}_{i,j'}, j \neq j', \text{ and } (v_{i,j,k} \leftrightarrow v_{i,j',k'}) \in \mathcal{E}'\}$. This set includes edges $(v_{i,j,k} \leftrightarrow v_{i,j',k'})$ that connect nodes $v_{i,j,k}$ and $v_{i,j',k'}$ from different visit-subgraphs $G_{i,j}$ and $G_{i,j'}$ respectively, provided that there exists an edge $(v_{i,j,k} \leftrightarrow v_{i,j',k'})$ in the global graph $G'$. The final representation of the patient's personalized KG, $G_{\text{pat}(i)}$, integrating both the visit-specific data and the broader inter-visit connections, is given as: $G_{\text{pat}(i)} = \left(\mathcal{P} \cup \bigcup_{j=1}^J \mathcal{V}'_{i,j}, \epsilon \cup \left(\bigcup_{j=1}^J \mathcal{E}'_{i,j}\right) \cup \mathcal{E}_{\text{inter}}\right)$.

## 3.3 STEP 3: **B**I-ATTENTION **A**UGMEN**T**ED GRAPH NEURAL NETWORK

Given that each patient's data encompasses multiple visit-subgraphs, it becomes imperative to devise a specialized model capable of managing this temporal graph data. Graph Neural Networks (GNNs), known for their proficiency in this domain, can be generalized as:

$$\mathbf{h}_k^{(l+1)} = \sigma\left(\mathbf{W}^{(l)}\text{AGGREGATE}^{(l)}\left(\mathbf{h}_{k'}^{(l)} | k' \in \mathcal{N}(k)\right) + \mathbf{b}^{(l)}\right), \tag{1}$$

where $\mathbf{h}_k^{(l+1)}$ represents the updated node representation of node $k$ at the $(l + 1)$-th layer of the GNN. The function $\text{AGGREGATE}^{(l)}$ aggregates the node representations of all neighbors $\mathcal{N}(k)$ of node $k$ at the $l$-th layer. $\mathbf{W}^{(l)}$ and $\mathbf{b}^{(l)}$ are the learnable weight matrix and bias vector at the $l$-th layer, respectively. $\sigma$ denotes an activation function. Nonetheless, the conventional GNN approach overlooks the temporal characteristics of our patient-specific graphs and misses the intricacies of personalized healthcare. To address this, we propose a Bi-attention Augmented (BAT) GNN that better accommodates temporal graph data and offers more nuanced predictive healthcare insights.

**Our model.** In GRAPHCARE, we incorporate attention mechanisms to effectively capture relevant information from the personalized KG. We first reduce the size of node and edge embedding from the word embedding to the hidden embedding to improve model's efficiency. The dimension-reduced hidden embeddings are computed as follows:

$$\mathbf{h}_{i,j,k} = \mathbf{W}_v \mathbf{h}_{(i,j,k)}^\mathcal{V} + \mathbf{b}_v \quad \mathbf{h}_{(i,j,k) \leftrightarrow (i,j',k')} = \mathbf{W}_r \mathbf{h}_{(i,j,k) \leftrightarrow (i,j',k')}^\mathcal{R} + \mathbf{b}_r \tag{2}$$

where $\mathbf{W}_v, \mathbf{W}_r \in \mathbb{R}^{w \times q}$, $\mathbf{b}_v, \mathbf{b}_r \in \mathbb{R}^q$ are learnable vectors, $\mathbf{h}_{(i,j,k)}^{\mathcal{V}}, \mathbf{h}_{(i,j,k)\leftrightarrow(i,j',k')}^{\mathcal{R}} \in \mathbb{R}^w$ are input embedding, $\mathbf{h}_{i,j,k}, \mathbf{h}_{(i,j,k)\leftrightarrow(i,j',k')} \in \mathbb{R}^q$ are hidden embedding of the $k$-th node in $j$-th visit-subgraph of patient, and the hidden embedding of the edge between the nodes $v_{i,j,k}$ and $v_{i,j'k'}$, respectively. $q$ is the size of the hidden embedding.

Subsequently, we compute two sets of attention weights: one set corresponds to the subgraph associated with each visit, and the other pertains to the nodes within each subgraph. The node-level attention weight for the $k$-th node in the $j$-th visit-subgraph of patient $i$, denoted as $\alpha_{i,j,k}$, and the visit-level attention weight for the $j$-th visit of patient $i$, denoted as $\beta_{i,j}$, are shown as follows:

$$\alpha_{i,j,1}, ..., \alpha_{i,j,M} = \text{Softmax}(\mathbf{W}_\alpha \mathbf{g}_{i,j} + \mathbf{b}_\alpha),$$

$$\beta_{i,1}, ..., \beta_{i,N} = \boldsymbol{\lambda}^\top \text{Tanh}(\mathbf{w}_\beta^\top \mathbf{G}_i + \mathbf{b}_\beta), \quad \text{where} \quad \boldsymbol{\lambda} = [\lambda_1, ..., \lambda_N], \tag{3}$$

where $\mathbf{g}_{i,j} \in \mathbb{R}^M$ is a multi-hot vector representation of visit-subgraph $G_{i,j}$, indicating the nodes appeared for the $j$-th visit of patient $i$ where $M = |\mathcal{V}'|$ is the number of nodes in the global graph $G'$. $\mathbf{G}_i \in \mathbb{R}^{N \times M}$ represents the multi-hot matrix of patient $i$'s graph $G_i$ where $N$ is the maximum visits across all patients. $\mathbf{W}_\alpha \in \mathbb{R}^{M \times M}$, $\mathbf{w}_\beta \in \mathbb{R}^M$, $\mathbf{b}_\alpha \in \mathbb{R}^M$ and $\mathbf{b}_\beta \in \mathbb{R}^N$ are learnable parameters. $\boldsymbol{\lambda} \in \mathbb{R}^N$ is the decay coefficient vector, $J$ is the number of visits of patient $i$, $\lambda_j \in \boldsymbol{\lambda}$ where $\lambda_j = \exp(-\gamma(J - j))$ when $j \leq J$ and 0 otherwise, is coefficient for the visit $j$, with decay rate $\gamma$, initializing higher weights for more recent visits.

*Attention initialization.* To further incorporate prior knowledge from LLMs and help the model converge, we initialize $\mathbf{W}_\alpha$ for the node-level attention based on the cosine similarity between the node embedding and the word embedding $\mathbf{w}_{\text{tf}}$ of a specific term for the a prediction task-feature pair (e.g., "terminal condition" for mortality-condition). We provide more details on this in Appendix C. Formally, we first calculate the weights for the nodes in the global graph $G'$ by $w_m = (\mathbf{h}_m \cdot \mathbf{w}_{\text{tf}})/(||\mathbf{h}_m||_2 \cdot ||\mathbf{w}_{\text{tf}}||_2)$ where $\mathbf{h}_m \in \mathbf{H}^{\mathcal{V}}$ is the input embedding of the $m$-th node in $G'$, and $w_m$ is the weight computed. We normalize the weights s.t. $0 \leq w_m \leq 1, \forall 1 \leq m \leq M$. We initialize $\mathbf{W}_\alpha = \text{diag}(w_1, ..., w_M)$ as a diagonal matrix.

Next, we update the node embedding by aggregating the neighboring nodes across all visit-subgraphs incorporating the attention weights for visits and nodes computed in Eq (3) and the weights for edges. Based on Eq (1), we design our convolutional layer BAT as follows:

$$\mathbf{h}_{i,j,k}^{(l+1)} = \sigma\left(\mathbf{W}^{(l)} \sum_{j' \in J, k' \in \mathcal{N}(k) \cup \{k\}} \left(\underbrace{\alpha_{i,j',k'}^{(l)} \beta_{i,j'}^{(l)} \mathbf{h}_{i,j',k'}^{(l)}}_{\text{Node aggregation term}} + \underbrace{\mathrm{w}_{\mathcal{R}\langle k,k'\rangle}^{(l)} \mathbf{h}_{(i,j,k)\leftrightarrow(i,j',k')}}_{\text{Edge aggregation term}}\right) + \mathbf{b}^{(l)}\right),$$
$$\tag{4}$$

where $\sigma$ is the ReLU function, $\mathbf{W}^{(l)} \in \mathbb{R}^{q \times q}, \mathbf{b}^{(l)} \in \mathbb{R}^q$ are learnable parameters, $\mathbf{w}_{\mathcal{R}}^{(l)} \in \mathbb{R}^{|\mathcal{E}'|}$ is the edge weight vector at the layer $l$, and $\mathrm{w}_{\mathcal{R}\langle k,k'\rangle}^{(l)} \in \mathbf{w}_{\mathcal{R}}^{(l)}$ is the scalar weight for the edge embedding $\mathbf{h}_{(i,j,k)\leftrightarrow(i,j',k')}^{\mathcal{R}}$. In Eq (4), the node aggregation term captures the contribution of the attention-weighted nodes, while the edge aggregation term represents the influence of the edges connecting the nodes. This convolutional layer integrates both node and edge features, allowing the model to learn a rich representation of the patient's EHR data. After several layers of convolution, we obtain the node embeddings $\mathbf{h}_{i,j,k}^{(L)}$ of the final layer ($L$), which are used for the predictions:

$$\mathbf{h}_i^{G_{\text{pat}}} = \text{MEAN}(\sum_{j=1}^{J} \sum_{k=1}^{K_j} \mathbf{h}_{i,j,k}^{(L)}), \quad \mathbf{h}_i^{\mathcal{P}} = \text{MEAN}(\sum_{j=1}^{J} \sum_{k=1}^{K_j} \mathbb{1}_{i,j,k}^{\Delta} \mathbf{h}_{i,j,k}^{(L)}),$$

$$\mathbf{z}_i^{\text{graph}} = \text{MLP}(\mathbf{h}_i^{G_{\text{pat}}}), \quad \mathbf{z}_i^{\text{node}} = \text{MLP}(\mathbf{h}_i^{\mathcal{P}}), \quad \mathbf{z}_i^{\text{joint}} = \text{MLP}(\mathbf{h}_i^{G_{\text{pat}}} \oplus \mathbf{h}_i^{\mathcal{P}}), \tag{5}$$

where $J$ is the number of visits of patient $i$, $K_j$ is the number of nodes in visit $j$, $\mathbf{h}_i^{G_{\text{pat}}}$ denotes the patient graph embedding obtained by averaging the embeddings of all nodes across visit-subgraphs and the various nodes within each subgraph for patient $i$. $\mathbf{h}_i^{\mathcal{P}}$ represents the patient node embedding computed by averaging node embeddings of the direct medical concept linked to the patient node. $\mathbb{1}_{i,j,k}^{\Delta} \in \{0, 1\}$ is a binary label indicating whether a node $v_{i,j,k}$ corresponds to a direct medical concept for patient $i$. Finally, we apply a multilayer perception (MLP) to either $\mathbf{h}_i^{G_{\text{pat}}}$, $\mathbf{h}_i^{\mathcal{P}}$, or the concatenated embedding ($\mathbf{h}_i^{G_{\text{pat}}} \oplus \mathbf{h}_i^{\mathcal{P}}$) to obtain logits $\mathbf{z}_i^{\text{graph}}$, $\mathbf{z}_i^{\text{node}}$ or $\mathbf{z}_i^{\text{joint}}$ respectively. We discuss more details of the patient representation learning in Appendix E.

Table 1: Statistics of pre-processed EHR datasets. "#": "the number of", "/patient": "per patient".

|  | #patients | #visits | #visits/patient | #conditions/patient | #procedures/patient | #drugs/patient |
|---|---|---|---|---|---|---|
| MIMIC-III | 35,707 | 44,399 | 1.24 | 12.89 | 4.54 | 33.71 |
| MIMIC-IV | 123,488 | 232,263 | 1.88 | 21.74 | 4.70 | 43.89 |

## 3.4 TRAINING AND PREDICTION

The model can be adapted for a variety of healthcare prediction tasks. Consider a set of samples $\{(x_1), (x_1, x_2), \ldots, (x_1, x_2, \ldots, x_t)\}$ for each patient with $t$ visits, where each tuple represents a sample consisting of a sequence of consecutive visits.

**Mortality (MT.) prediction** predicts the mortality label of the subsequent visit for each sample, with the last sample dropped. Formally, $f : (x_1, x_2, \ldots, x_{t-1}) \rightarrow y[x_t]$ where $y[x_t] \in \{0, 1\}$ is a binary label indicating the patient's survival status recorded in visit $x_t$.

**Readmission (RA.) prediction** predicts if the patient will be readmitted into hospital within $\sigma$ days. Formally, $f : (x_1, x_2, \ldots, x_{t-1}) \rightarrow y[\tau(x_t) - \tau(x_{t-1})], y \in \{0, 1\}$ where $\tau(x_t)$ denotes the encounter time of visit $x_t$. $y[\tau(x_t) - \tau(x_{t-1})]$ equals 1 if $\tau(x_t) - \tau(x_{t-1}) \leq \sigma$, and 0 otherwise. In our study, we set $\sigma = 15$ days.

**Length-Of-Stay (LOS) prediction** (Harutyunyan et al., 2019) predicts the length of ICU stays for each visit. Formally, $f : (x_1, x_2, \ldots, x_t) \rightarrow y[x_t]$ where $y[x_t] \in \mathbb{R}^{1 \times C}$ is a one-hot vector indicating its class among $C$ classes. We set 10 classes $[\mathbf{0}, \mathbf{1}, \ldots, \mathbf{7}, \mathbf{8}, \mathbf{9}]$, which signify the stays of length $< 1$ day ($\mathbf{0}$), within one week ($\mathbf{1}, \ldots, \mathbf{7}$), one to two weeks ($\mathbf{8}$), and $\geq$ two weeks ($\mathbf{9}$).

**Drug recommendation** predicts medication labels for each visit. Formally, $f : (x_1, x_2, \ldots, x_t) \rightarrow y[x_t]$ where $y[x_t] \in \mathbb{R}^{1 \times |\mathbf{d}|}$ is a multi-hot vector where $|\mathbf{d}|$ denotes the number of all drug types.

We use binary cross-entropy (BCE) loss with sigmoid function to train binary (MT. and RA.) and multi-label classification (Drug.) classification tasks, while we use corss-entropy (CE) loss with softmax function to train multi-class (LOS) classification tasks.

## 4 EXPERIMENTS

### 4.1 EXPERIMENTAL SETTING

**Data**. For the EHR data, we use the publicly available MIMIC-III (Johnson et al., 2016) and MIMIC-IV (Johnson et al., 2020) datasets. Table 1 presents statistics of the processed datasets. To build concept-specific KG (§3.1), we utilize GPT-4 (OpenAI, 2023) as the LLM for KG generation, and utilize UMLS-KG (Bodenreider, 2004) as an existing biomedical KG for subgraph sampling, which has 300K entities and 1M relations. $\chi = 3$ and $\kappa = 1$ are set as parameters. We employ the GPT-3 embedding model to retrieve the word embeddings of the entities and relations.

**Baselines.** Our baselines include GRU (Chung et al., 2014), Transformer (Vaswani et al., 2017), RETAIN (Choi et al., 2016c), GRAM (Choi et al., 2017), Deepr (Nguyen et al., 2016), StageNet (Gao et al., 2020), AdaCare (Ma et al., 2020a), GRASP (Zhang et al., 2021b), SafeDrug (Yang et al., 2021b), MICRON (Yang et al., 2021a), GAMENet (Shang et al., 2019b), and MoleRec (Yang et al., 2023b). AdaCare and GRASP are evaluated only on binary prediction tasks given their computational demands. For drug recommendation, we also consider task-specific models SafeDrug, MICRON, GAMENet, and MoleRec. Our GRAPHCARE model's performance is examined under five GNNs and graph transformers: GAT (Veličković et al., 2017), GINE (Hu et al., 2019), EGT (Hussain et al., 2022), GPS (Rampášek et al., 2022) and our BAT. We do not compare to models such as GCT (Choi et al., 2020) and CGL (Lu et al., 2021a) as they incorporate lab results and clinical notes, which are not used in this study. Implementation details are discussed in Appendix C.

**Evaluation Metrics.** We consider the following metrics: (a) **Accuracy** - the proportion of correctly predicted instances out of the total instances; (b) **F1** - the harmonic mean of precision and recall; (c) **Jaccard score** - the ratio of the intersection to the union of predicted and true labels; (d) **AUPRC** - the area under the precision-recall curve, emphasizing the trade-off between precision and recall; (e) **AUROC** - the area under the receiver operating characteristic curve, capturing the trade-off between the true positive and the false positive rates. (f) **Cohen's Kappa** - measures inter-rater agreement for categorical items, adjusting for the expected level of agreement by chance in multi-class classification.

Table 2: **Performance comparison of four prediction tasks on MIMIC-III/MIMIC-IV.** We report the average performance (%) and the standard deviation (in bracket) of each model over 100 runs for MIMIC-III and 25 runs for MIMIC-IV. The best results are **highlighted** for both datasets.

| Model | | **Task 1: Mortality Prediction** | | | | **Task 2: Readmission Prediction** | | | |
|---|---|---|---|---|---|---|---|---|---|
| | | **MIMIC-III** | | **MIMIC-IV** | | **MIMIC-III** | | **MIMIC-IV** | |
| | | **AUPRC** | **AUROC** | **AUPRC** | **AUROC** | **AUPRC** | **AUROC** | **AUPRC** | **AUROC** |
| GRU | | $11.8_{(0.5)}$ | $61.3_{(0.9)}$ | $4.2_{(0.1)}$ | $69.0_{(0.8)}$ | $68.2_{(0.4)}$ | $65.4_{(0.8)}$ | $66.1_{(0.1)}$ | $66.2_{(0.1)}$ |
| Transformer | | $10.1_{(0.9)}$ | $57.2_{(1.3)}$ | $3.4_{(0.4)}$ | $65.1_{(1.2)}$ | $67.3_{(0.7)}$ | $63.9_{(1.1)}$ | $65.7_{(0.3)}$ | $65.3_{(0.4)}$ |
| RETAIN | | $9.6_{(0.6)}$ | $59.4_{(1.5)}$ | $3.8_{(0.4)}$ | $64.8_{(1.6)}$ | $65.1_{(1.0)}$ | $64.1_{(0.7)}$ | $66.2_{(0.3)}$ | $66.3_{(0.2)}$ |
| GRAM | | $11.4_{(0.7)}$ | $60.4_{(0.9)}$ | $4.4_{(0.3)}$ | $66.7_{(0.7)}$ | $67.2_{(0.8)}$ | $64.3_{(0.4)}$ | $66.1_{(0.2)}$ | $66.3_{(0.3)}$ |
| Deepr | | $13.2_{(1.1)}$ | $60.8_{(0.4)}$ | $4.2_{(0.2)}$ | $68.9_{(0.9)}$ | $68.8_{(0.9)}$ | $66.5_{(0.4)}$ | $65.6_{(0.1)}$ | $65.4_{(0.2)}$ |
| AdaCare | | $11.1_{(0.4)}$ | $58.4_{(1.4)}$ | $4.6_{(0.3)}$ | $69.3_{(0.7)}$ | $68.6_{(0.6)}$ | $65.7_{(0.3)}$ | $65.9_{(0.0)}$ | $66.1_{(0.0)}$ |
| GRASP | | $9.9_{(1.1)}$ | $59.2_{(1.4)}$ | $4.7_{(0.1)}$ | $68.4_{(1.0)}$ | $69.2_{(0.4)}$ | $66.3_{(0.6)}$ | $66.3_{(0.3)}$ | $66.1_{(0.2)}$ |
| StageNet | | $12.4_{(0.3)}$ | $61.5_{(0.7)}$ | $4.2_{(0.3)}$ | $69.6_{(0.8)}$ | $69.3_{(0.6)}$ | $66.7_{(0.4)}$ | $66.1_{(0.1)}$ | $66.2_{(0.1)}$ |
| GRAPHCARE | w/ GAT | $14.3_{(0.8)}$ | $67.8_{(1.1)}$ | $5.1_{(0.1)}$ | $71.8_{(1.0)}$ | $71.5_{(0.7)}$ | $68.1_{(0.6)}$ | $67.4_{(0.4)}$ | $67.3_{(0.4)}$ |
| | w/ GINE | $14.4_{(0.4)}$ | $67.3_{(1.3)}$ | $5.7_{(0.1)}$ | $72.0_{(1.1)}$ | $71.3_{(0.8)}$ | $68.0_{(0.4)}$ | $68.3_{(0.3)}$ | $67.5_{(0.4)}$ |
| | w/ EGT | $15.5_{(0.5)}$ | $69.1_{(1.0)}$ | $6.2_{(0.2)}$ | $71.3_{(0.7)}$ | $72.2_{(0.5)}$ | $68.8_{(0.4)}$ | $68.9_{(0.2)}$ | $67.6_{(0.3)}$ |
| | w/ GPS | $15.3_{(0.9)}$ | $68.8_{(0.8)}$ | $\mathbf{6.7_{(0.2)}}$ | $72.7_{(0.9)}$ | $71.9_{(0.6)}$ | $68.5_{(0.6)}$ | $69.1_{(0.4)}$ | $67.9_{(0.4)}$ |
| | w/ BAT | $\mathbf{16.7_{(0.5)}}$ | $\mathbf{70.3_{(0.5)}}$ | $\mathbf{6.7_{(0.3)}}$ | $\mathbf{73.1_{(0.5)}}$ | $\mathbf{73.4_{(0.4)}}$ | $\mathbf{69.7_{(0.5)}}$ | $\mathbf{69.6_{(0.3)}}$ | $\mathbf{68.5_{(0.4)}}$ |

| Model | | **Task 3: Length of Stay Prediction** | | | | | | | |
|---|---|---|---|---|---|---|---|---|---|
| | | **MIMIC-III** | | | | **MIMIC-IV** | | | |
| | | **AUROC** | **Kappa** | **Accuracy** | **F1-score** | **AUROC** | **Kappa** | **Accuracy** | **F1-score** |
| GRU | | $78.3_{(0.1)}$ | $26.2_{(0.2)}$ | $40.3_{(0.3)}$ | $34.9_{(0.5)}$ | $78.7_{(0.1)}$ | $26.0_{(0.1)}$ | $35.2_{(0.1)}$ | $31.6_{(0.2)}$ |
| Transformer | | $78.3_{(0.2)}$ | $25.4_{(0.4)}$ | $40.1_{(0.3)}$ | $34.8_{(0.2)}$ | $78.3_{(0.3)}$ | $25.3_{(0.4)}$ | $34.4_{(0.2)}$ | $31.4_{(0.3)}$ |
| RETAIN | | $78.2_{(0.1)}$ | $26.1_{(0.4)}$ | $40.6_{(0.3)}$ | $34.9_{(0.4)}$ | $78.9_{(0.3)}$ | $26.3_{(0.2)}$ | $35.7_{(0.2)}$ | $32.0_{(0.2)}$ |
| GRAM | | $78.2_{(0.1)}$ | $26.3_{(0.3)}$ | $40.4_{(0.4)}$ | $34.5_{(0.2)}$ | $78.8_{(0.2)}$ | $26.1_{(0.4)}$ | $35.4_{(0.2)}$ | $31.9_{(0.3)}$ |
| Deepr | | $77.9_{(0.1)}$ | $25.3_{(0.4)}$ | $40.1_{(0.6)}$ | $35.0_{(0.4)}$ | $79.5_{(0.3)}$ | $26.4_{(0.2)}$ | $35.8_{(0.3)}$ | $32.3_{(0.1)}$ |
| StageNet | | $78.3_{(0.2)}$ | $24.8_{(0.2)}$ | $39.9_{(0.2)}$ | $34.4_{(0.4)}$ | $79.2_{(0.3)}$ | $26.0_{(0.2)}$ | $35.0_{(0.2)}$ | $31.3_{(0.3)}$ |
| GRAPHCARE | w/ GAT | $79.4_{(0.3)}$ | $28.2_{(0.2)}$ | $41.9_{(0.2)}$ | $36.1_{(0.4)}$ | $80.3_{(0.3)}$ | $28.4_{(0.4)}$ | $36.2_{(0.1)}$ | $33.3_{(0.3)}$ |
| | w/ GINE | $79.2_{(0.2)}$ | $28.3_{(0.3)}$ | $41.5_{(0.3)}$ | $36.0_{(0.4)}$ | $79.9_{(0.2)}$ | $27.5_{(0.3)}$ | $36.3_{(0.3)}$ | $32.8_{(0.2)}$ |
| | w/ EGT | $80.3_{(0.3)}$ | $28.8_{(0.2)}$ | $42.8_{(0.4)}$ | $36.3_{(0.5)}$ | $80.5_{(0.2)}$ | $28.7_{(0.3)}$ | $36.7_{(0.2)}$ | $33.5_{(0.1)}$ |
| | w/ GPS | $80.9_{(0.3)}$ | $28.8_{(0.3)}$ | $43.0_{(0.3)}$ | $36.8_{(0.4)}$ | $80.7_{(0.3)}$ | $28.8_{(0.4)}$ | $36.7_{(0.3)}$ | $33.9_{(0.3)}$ |
| | w/ BAT | $\mathbf{81.4_{(0.3)}}$ | $\mathbf{29.5_{(0.4)}}$ | $\mathbf{43.2_{(0.4)}}$ | $\mathbf{37.5_{(0.2)}}$ | $\mathbf{81.7_{(0.2)}}$ | $\mathbf{29.8_{(0.3)}}$ | $\mathbf{37.3_{(0.3)}}$ | $\mathbf{34.2_{(0.3)}}$ |

| Model | | **Task 4: Drug Recommendation** | | | | | | | |
|---|---|---|---|---|---|---|---|---|---|
| | | **MIMIC-III** | | | | **MIMIC-IV** | | | |
| | | **AUPRC** | **AUROC** | **F1-score** | **Jaccard** | **AUPRC** | **AUROC** | **F1-score** | **Jaccard** |
| GRU | | $77.0_{(0.1)}$ | $94.4_{(0.0)}$ | $62.3_{(0.3)}$ | $47.8_{(0.3)}$ | $74.1_{(0.1)}$ | $94.2_{(0.1)}$ | $60.2_{(0.2)}$ | $44.0_{(0.4)}$ |
| Transformer | | $76.1_{(0.1)}$ | $94.2_{(0.0)}$ | $62.1_{(0.4)}$ | $47.1_{(0.4)}$ | $71.3_{(0.1)}$ | $93.4_{(0.1)}$ | $55.9_{(0.2)}$ | $40.4_{(0.1)}$ |
| RETAIN | | $77.1_{(0.1)}$ | $94.4_{(0.0)}$ | $63.7_{(0.2)}$ | $48.8_{(0.2)}$ | $74.2_{(0.1)}$ | $94.3_{(0.0)}$ | $60.3_{(0.1)}$ | $45.0_{(0.1)}$ |
| GRAM | | $76.7_{(0.1)}$ | $94.2_{(0.1)}$ | $62.9_{(0.3)}$ | $47.9_{(0.3)}$ | $74.3_{(0.2)}$ | $94.2_{(0.1)}$ | $60.1_{(0.2)}$ | $45.3_{(0.3)}$ |
| Deepr | | $74.3_{(0.1)}$ | $93.7_{(0.0)}$ | $60.3_{(0.4)}$ | $44.7_{(0.3)}$ | $73.7_{(0.1)}$ | $94.2_{(0.1)}$ | $59.1_{(0.4)}$ | $43.8_{(0.4)}$ |
| StageNet | | $74.4_{(0.1)}$ | $93.0_{(0.1)}$ | $61.4_{(0.3)}$ | $45.8_{(0.4)}$ | $74.4_{(0.1)}$ | $94.2_{(0.0)}$ | $60.2_{(0.3)}$ | $45.4_{(0.4)}$ |
| SafeDrug | | $68.1_{(0.3)}$ | $91.0_{(0.1)}$ | $46.7_{(0.4)}$ | $31.7_{(0.3)}$ | $66.4_{(0.5)}$ | $91.8_{(0.2)}$ | $56.2_{(0.4)}$ | $44.3_{(0.3)}$ |
| MICRON | | $77.4_{(0.0)}$ | $94.6_{(0.1)}$ | $63.2_{(0.4)}$ | $48.3_{(0.4)}$ | $74.4_{(0.1)}$ | $94.3_{(0.1)}$ | $59.3_{(0.3)}$ | $44.1_{(0.3)}$ |
| GAMENet | | $76.4_{(0.0)}$ | $94.2_{(0.1)}$ | $62.1_{(0.4)}$ | $47.2_{(0.4)}$ | $74.2_{(0.1)}$ | $94.2_{(0.1)}$ | $60.4_{(0.4)}$ | $45.3_{(0.3)}$ |
| MoleRec | | $69.8_{(0.1)}$ | $92.0_{(0.1)}$ | $58.1_{(0.1)}$ | $43.1_{(0.3)}$ | $68.6_{(0.1)}$ | $92.1_{(0.1)}$ | $56.3_{(0.4)}$ | $40.6_{(0.3)}$ |
| GRAPHCARE | w/ GAT | $78.5_{(0.2)}$ | $94.8_{(0.1)}$ | $64.4_{(0.3)}$ | $49.2_{(0.4)}$ | $74.7_{(0.5)}$ | $94.4_{(0.3)}$ | $60.4_{(0.3)}$ | $45.7_{(0.4)}$ |
| | w/ GINE | $78.2_{(0.1)}$ | $94.7_{(0.1)}$ | $63.6_{(0.4)}$ | $47.9_{(0.3)}$ | $74.8_{(0.3)}$ | $94.6_{(0.1)}$ | $60.6_{(0.4)}$ | $46.1_{(0.4)}$ |
| | w/ EGT | $79.6_{(0.2)}$ | $95.3_{(0.0)}$ | $66.4_{(0.2)}$ | $49.6_{(0.4)}$ | $75.4_{(0.4)}$ | $95.0_{(0.1)}$ | $61.6_{(0.3)}$ | $47.3_{(0.3)}$ |
| | w/ GPS | $79.9_{(0.3)}$ | $\mathbf{95.5_{(0.1)}}$ | $66.2_{(0.3)}$ | $\mathbf{49.8_{(0.4)}}$ | $75.9_{(0.5)}$ | $94.9_{(0.1)}$ | $62.1_{(0.3)}$ | $46.8_{(0.4)}$ |
| | w/ BAT | $\mathbf{80.2_{(0.2)}}$ | $\mathbf{95.5_{(0.1)}}$ | $\mathbf{66.8_{(0.2)}}$ | $49.7_{(0.3)}$ | $\mathbf{77.1_{(0.1)}}$ | $\mathbf{95.4_{(0.2)}}$ | $\mathbf{63.9_{(0.3)}}$ | $\mathbf{48.1_{(0.3)}}$ |

## 4.2 EXPERIMENTAL RESULTS

As demonstrated in Table 2, GRAPHCARE consistently outperforms existing baselines across all prediction tasks for both MIMIC-III and MIMIC-IV datasets. For example, when combined with BAT, GRAPHCARE exceeds StageNet's best result by +14.3% in AUROC for mortality prediction on MIMIC-III. Within our GRAPHCARE framework, our proposed BAT GNN consistently performs the best, underlining the effectiveness of the bi-attention mechanism. In the following, we analyze the effects of incorporating the personalized KG and our proposed BAT in detail.

### 4.2.1 EFFECT OF PERSONALIZED KNOWLEDGE GRAPH

**Effect of EHR Data Size.** To examine the impact of training data volume on model performance, we conduct a comprehensive experiment where the size of the training data fluctuates between 0.1% and 100% of the original training set, while the validation/testing data remain constant.

Performance metrics are averaged over 10 runs, each initiated with a different random seed. The results, depicted in Figure 2, suggest that GRAPHCARE exhibits a considerable edge over other models when confronted with scarce training data. For instance, GRAPH-CARE, despite being trained on a mere 0.1% of the training data (36 patient samples), accomplished an LOS prediction accuracy comparable to the best baseline StageNet that trained on 2.0% of the training data (about 720 patient

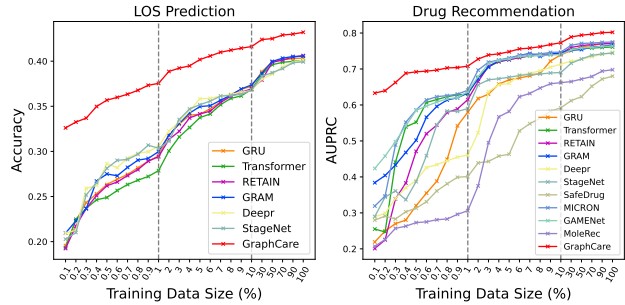

Figure 2: **Performance by EHR training data sizes**. Values on the x-axis indicate % of the entire training data. The dotted lines separate three ranges: [0.1, 1], [1, 10] and [10, 100] (%).

samples). A similar pattern appears in drug recommendation tasks. Notably, both GAMENet and GRAM also show a certain level of resilience against data limitations, likely due to their use of internal EHR graphs or external ontologies.

**Effect of Knowledge Graph Size.** Figure 3 illuminates how varying sizes of KGs influence the efficacy of GRAPHCARE. We test GPT-KG (generated by GPT-4), UMLS-KG (sampled from UMLS), and GPT-UMLS-KG (a combination). Key observations include: (1) Across all KGs, as the size ratio of the KG increases, there is a corresponding uptick in GRAPHCARE's performance. (2) The amalgamated GPT-UMLS-KG consistently outperforms the other two KGs. This underscores the premise that richer knowledge bases enable more precise clinical predictions. Moreover, it demonstrates GPT-KG and UMLS-KG could enrich each other with unseen knowledge. (3) The degree of KG contribution varies depending on the task at hand. Specifically, GPT-KG demonstrates a stronger influence over mortality and LOS predictions compared to UMLS-KG. Conversely, UMLS-KG edges out in readmission prediction, while both KGs showcase comparable capabilities in drug recommendations. (4) Notably, lower KG ratios (from 0.1 to 0.5) are associated with larger standard deviations, which is attributed to the reduced likelihood of vital knowledge being encompassed within the sparsely sampled sub-KGs.

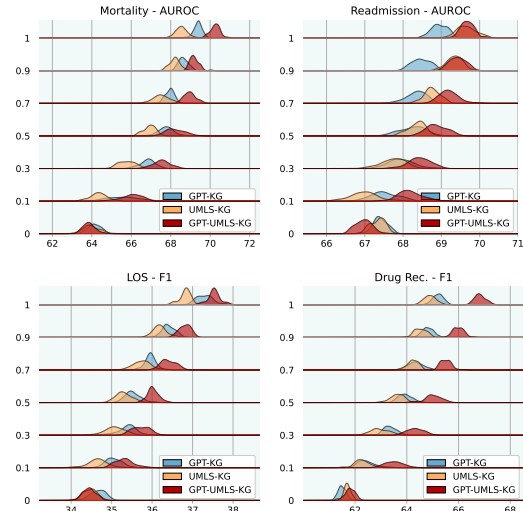

Figure 3: **Performance by different KG sizes.** We test on three distinct KGs: GPT-KG, UMLS-KG, and GPT-UMLS-KG. For each, we sample sub-KGs using varying ratios: $[0.0, 0.1, 0.3, 0.5, 0.7, 0.9, 1.0]$ while ensuring the nodes corresponding to EHR medical concepts remain consistent across samples. The distributions are based on 30 runs on the MIMIC-III with different random seeds.

### 4.2.2 EFFECT OF BI-ATTENTION AUGMENTED GRAPH NEURAL NETWORK

Table 3 provides an in-depth ablation study on the proposed GNN BAT, highlighting the profound influence of distinct components on the model's effectiveness.

The data reveals that excluding node-level attention ($\alpha$) results in a general drop in performance across tasks for both datasets. This downturn is particularly pronounced for the drug recommendation task. Regarding visit-level attention ($\beta$), the effects of its absence are more discernible in the MIMIC-IV dataset. This is likely attributed to MIMIC-IV's larger average number of visits per patient, as outlined in Table 1. Given this disparity, the ability to discern between distinct visits becomes pivotal across all tasks. Moreover, when considering tasks, it's evident that the RA. task is particularly vulnerable to adjustments in visit-level attention ($\beta$) and edge weight ($w_{\mathcal{R}}$). This underlines the significance of capturing visit-level nuances and inter-entity relationships within the EHR to ensure precise RA. outcome predictions. Regarding attention initialization (*AttnInit*), it emerges as a crucial factor in priming the model to be more receptive to relevant clinical insights from the get-go. Omitting this initialization shows a noticeable decrement in performance, particularly for

Table 3: **Variant analysis of BAT.** We measure AUROC for MT. and RA. prediction, and F1-score for the tasks of LOS prediction and drug recommendation. $\alpha$, $\beta$, $w_{\mathcal{R}}$, and *AttnInit* are node-level, visit-level attention, edge weight, and attention initialization, respectively. We report the average performance of 10 runs for each case.

| Case | Variants | MIMIC-III | | | | MIMIC-IV | | | |
| | | MT. | RA. | LOS | Drug. | MT. | RA. | LOS | Drug. |
| --- | --- | --- | --- | --- | --- | --- | --- | --- | --- |
| #0 | w/ *all* | 70.3 | 69.7 | 37.5 | 66.8 | 73.1 | 68.5 | 34.2 | 63.9 |
| #1 | w/o $\alpha$ | $68.7_{\downarrow 0.6}$ | $68.5_{\downarrow 1.2}$ | $36.7_{\downarrow 0.8}$ | $64.6_{\downarrow 2.2}$ | $72.2_{\downarrow 0.9}$ | $67.8_{\downarrow 0.7}$ | $33.1_{\downarrow 1.1}$ | $61.6_{\downarrow 2.3}$ |
| #2 | w/o $\beta$ | $69.9_{\downarrow 0.4}$ | $68.7_{\downarrow 1.0}$ | $37.2_{\downarrow 0.3}$ | $66.5_{\downarrow 0.3}$ | $72.1_{\downarrow 1.0}$ | $67.0_{\downarrow 1.5}$ | $33.5_{\downarrow 0.7}$ | $63.2_{\downarrow 0.7}$ |
| #3 | w/o $w_{\mathcal{R}}$ | $69.8_{\downarrow 0.5}$ | $68.4_{\downarrow 1.3}$ | $36.8_{\downarrow 0.7}$ | $66.3_{\downarrow 0.5}$ | $72.9_{\downarrow 0.2}$ | $67.9_{\downarrow 0.6}$ | $33.7_{\downarrow 0.5}$ | $63.1_{\downarrow 0.8}$ |
| #4 | w/o *AttnInit* | $69.5_{\downarrow 0.8}$ | $69.2_{\downarrow 0.5}$ | $37.2_{\downarrow 0.3}$ | $65.5_{\downarrow 1.3}$ | $72.5_{\downarrow 0.6}$ | $68.1_{\downarrow 0.4}$ | $34.1_{\downarrow 0.1}$ | $62.4_{\downarrow 1.5}$ |
| #5 | w/o #(1,2,3,4) | $67.4_{\downarrow 2.9}$ | $68.1_{\downarrow 1.6}$ | $36.0_{\downarrow 1.5}$ | $64.0_{\downarrow 2.8}$ | $71.7_{\downarrow 1.4}$ | $67.5_{\downarrow 1.0}$ | $32.9_{\downarrow 1.3}$ | $60.5_{\downarrow 3.4}$ |

drug recommendations. This suggests that by guiding initial attention towards potentially influential nodes in the personalized KG, the model can more adeptly assimilate significant patterns and make informed predictions.

### 4.3 INTERPRETABILITY OF GRAPHCARE.

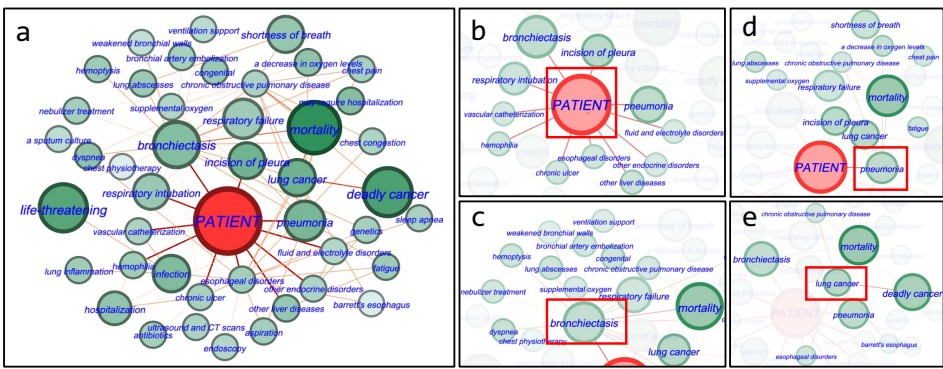

Figure 4: Example showing a patient's personalized KG with importance scores (Appendix F) visualized. For better presentation, we hide the nodes of drugs. The red node represents the patient node. Nodes with higher scores are darker and larger. Edges with higher scores are darker and thicker. Subgraph (a) shows a central area overview of this personalized KG, and other subgraphs show more details with a focused node highlighted.

Figure 4 showcases an example of a personalized KG for mortality prediction tied to a specific patient (predicted mortality 1), who was accurately predicted only by our GRAPHCARE method, while other baselines incorrectly estimated the outcome. In Figure 4a, important nodes and edges contributing to mortality prediction, such as "*deadly cancer*", are emphasized with higher importance scores. This demonstrates the effectiveness of our BAT model in identifying relevant nodes and edges. Additionally, Figure 4b shows the direct EHR nodes connected to the patient node, enhancing interpretability of predictions using patient node embedding. Figure 4c and 4d show KG triples linked to the direct EHR nodes "*bronchiectasis*" and "*pneumonia*". These nodes are connected to important nodes like "*mortality*", "*respiratory failure*", "*lung cancer*", and "*shortness of breath*", indicating their higher weights. In Figure 4e, the "*lung cancer*" node serves as a common connector for "*bronchiectasis*" and "*pneumonia*". It is linked to both "*mortality*" and "*deadly cancer*", highlighting its significance. Removing this node had a noticeable impact on the model's performance, indicating its pivotal role in accurate predictions. This emphasizes the value of comprehensive health data and considering all potential health factors, no matter how indirectly connected they may seem.

## 5 CONCLUSION

We presented GRAPHCARE, a framework that builds personalized knowledge graphs for enhanced healthcare predictions. Empirical studies show its dominance over baselines in various tasks on two datasets. With its robustness to limited data and scalability with KG size, GRAPHCARE promises significant potential in healthcare. We discuss ethics, limitations, and risks in Appendix A. Our code is available at https://github.com/pat-jj/GraphCare.

## 6 ACKNOWLEDGMENTS

This work was supported by NSF award SCH-2205289, SCH-2014438, and IIS-2034479.

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

## Contents of Appendix

# A  ETHICS, LIMITATIONS, AND RISKS

In this study, we introduce a novel framework, GRAPHCARE, which generates knowledge graphs (KGs) by leveraging relational knowledge from large language models (LLMs) and extracting information from existing KGs. This methodology is designed to provide an advanced tool for healthcare prediction tasks, enhancing their accuracy and interpretability. However, the ethical considerations associated with our approach warrant careful attention. Previous research has shown that LLMs may encode biases related to race, gender, and other demographic attributes (Sheng et al., 2020; Weidinger et al., 2021). Furthermore, they may potentially generate toxic outputs (Gehman et al., 2020). Such biases and toxicity could inadvertently influence the content of the knowledge graphs generated by our proposed GRAPHCARE, which relies on these LLMs for information extraction. Furthermore, the issue of privacy has emerged as a paramount concern associated with LLM usage (Lund & Wang, 2023).

We explain the limitations of GRAPHCARE and describe the measures we have implemented to counteract or mitigate these ethical concerns as follows.

## A.1  PREVENTING TOXIC BEHAVIORS AND ENSURING PATIENT PRIVACY

Primarily, the LLM within GRAPHCARE is exclusively utilized to extract knowledge associated with medical concepts. This focused usage drastically reduces the chances of inheriting wider social biases or manifesting toxic behaviors intrinsic to the parent LLMs. Furthermore, we ensure that no patient data is introduced into any open-source software. This measure fortifies patient confidentiality and negates the possibility of injecting individual biases into the knowledge graphs. This commitment is further elucidated in Figure 5.

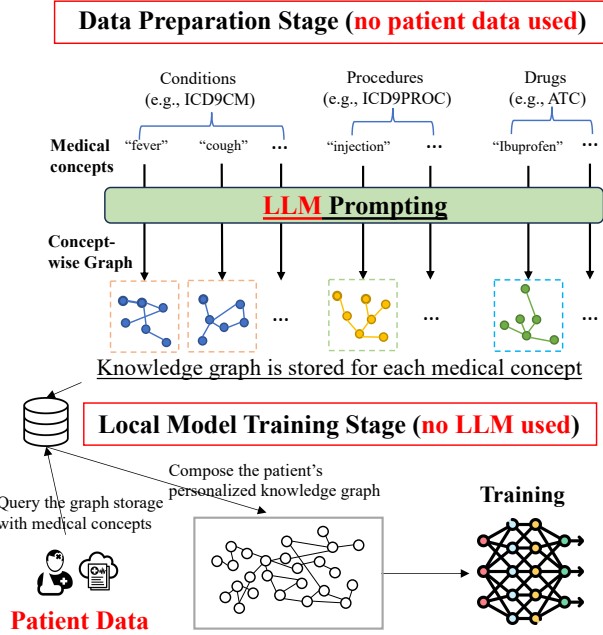

Figure 5: **High-level View of GRAPHCARE for Clarification on Ethical Considerations.** GRAPHCARE consists of two general stages: data preparation and local model training. During data preparation, the LLM solely extracts knowledge graphs associated with medical concepts, without accessing any patient's data. At the local model training stage, personalized knowledge graphs for patients are constructed using the knowledge graphs corresponding to medical concepts found in the patient's EHR, without any engagement of the LLM. A local graph storage serves as both the repository for the procured medical concept-wise KGs and the mechanism for querying KGs for personalized KG compositions.

## A.2 Counteracting the Adverse Effects of LLM Hallucinations

The predictive efficacy of GRAPHCARE is intrinsically tied to the veracity of knowledge graph triples sourced from LLMs. Hence, hallucinations within LLMs can detrimentally skew the performance of the model. To counterbalance this, we collaborate with a medical professional (MD) to scrutinize the accuracy of LLM-derived triples and expunged content that might be detrimental (further details are provided in Appendix D.1). Leveraging domain expert knowledge on triple evaluation and selection greatly minimizes the negative impacts of LLM hallucinations, ensuring a high-quality knowledge probing from LLM.

## A.3 Application

It's important to emphasize that GRAPHCARE is primarily intended for **research purposes**. This means that while it offers insights and can provide valuable information, it has not been certified or endorsed for clinical or diagnostic use. Any implementation or interpretation of GRAPHCARE should be undertaken with the clear understanding of its experimental nature.

While GRAPHCARE serves as an advanced tool for healthcare prediction tasks, it should not replace or undermine the expertise of medical professionals. We strongly advise against relying solely on its predictions for healthcare decisions. Medical doctors possess extensive training and clinical experience, and their judgment should always be prioritized over automated systems. Patients and healthcare providers should use the information from GRAPHCARE as supplementary and should always consult with healthcare professionals before making any medical decisions.

## B  EHR Dataset Processing

In this paper, we use MIMIC-III and MIMIC-IV datasets. Both datasets are under PhysioNet Credentialed Health Data License 1.5.0[2] We employ PyHealth (Yang et al., 2023a) to process these two datasets. PyHealth has an EHR dataset pre-processing pipeline that standardizes the datasets, organizing each patient's data into several visits, where each visit contains unique and specified feature lists. For our experiments, we create feature lists composed of conditions and procedures for Length of Stay (LOS) prediction and drug recommendations. For the prediction tasks of mortality and readmission, we include the medication (drug) list in addition to the condition and procedure lists.

Subsequent to the parsing of the datasets, PyHealth also enables the mapping of medical concepts across various coding systems using the provided code maps. The involved coding systems in this process are ICD-9[3], ICD-10[4,5], CCS[6], NDC[7] and ATC[8]. In our experiment, we convert 11,736 ICD-9-CM codes and 72,446 ICD-10-CM codes into 285 CCS-CM codes to capture condition concepts. Similarly, we map 4,670 ICD-9-PROC codes (a part of ICD-9-CM for procedure coding) and 79,758 ICD-10-PCS codes to 231 CCS-PROC codes for procedure concepts. For drug concepts, we convert 1,143,020 NDC codes into 269 level-3 ATC codes. This mapping process enhances the training speed and predictive performance of the model by reducing the granularity of medical concepts.

## C  Implementation Details

In this section, we present the implementation details of GRAPHCARE, aligning it closely with the methodology described in Section 3, which improves reproducibility and clarity.

---

[2]https://physionet.org/content/mimiciii/view-license/1.4/
[3]https://www.cdc.gov/nchs/icd/icd9cm.htm
[4]https://www.cms.gov/medicare/icd-10/2023-icd-10-cm
[5]https://www.cms.gov/medicare/icd-10/2023-icd-10-pcs
[6]https://www.hcup-us.ahrq.gov/toolssoftware/ccs/ccs.jsp
[7]https://www.accessdata.fda.gov/scripts/cder/ndc/index.cfm
[8]https://www.who.int/tools/atc-ddd-toolkit/atc-classification

## C.1 Implementation Details of Step 1 (§3.1): Concept-Specific KG Generation

**Medical concepts c, p, and d.** After the EHR data preprocessing illustrated in Appendix B, we have 285 conditions ($|\mathbf{c}| = 285$), 231 procedures ($|\mathbf{p}| = 231$), and 269 drugs ($|\mathbf{d}| = 269$).

**LLM-based KG extraction.** We detail this process in Appendix D.1.

**Subgraph sampling from existing KGs.** We detail this process in Appendix D.2.

The hyperparameter studies regarding (1) LLM prompting times $\chi$, and (2) $\kappa$ hops from the source entity are showcased in Table 6, 7, and 8.

**Node and Edge Clustering.** To analyze the global graph $G$, we obtain the word embeddings for each node and edge. These embeddings have 1536 dimensions and are sourced from the second-generation GPT-3 model, specifically the `text-embedding-ada-002`[9]). The model will output a single vector embedding of the input text regardless of the number of tokens it contains. We use Scikit-learn 1.2.1 (Pedregosa et al., 2011) to implement agglomerative clustering. We detail the hyperparameter study (for distance threshold $\delta$) of clustering in Appendix G.1. $\delta = 0.15$ was chosen based on the study.

## C.2 Implementation Details of Step 2 (§3.2): Personalized KG Composition

**Visit Processing**: Iterate through each visit in the patient's EHR. For each visit $\text{visit}_j$, we have the medical concepts $\{\text{concept}_{j1}, \text{concept}_{j2}, ...\}$.

**Inter-Visit Relationships**: Identify and establish connections between nodes across different visits if they share a relationship in the global graph $G^{'}$. These connections are represented by the set $\mathcal{E}_{\text{inter}}$, highlighting the continuity and progression in the patient's medical history.

**Final Personalized KG Assembly**: Combine all the integrated concept-specific KGs, the patient node $\mathcal{P}$, and the inter-visit relationships to form the final personalized KG for the patient, $G_{\text{pat}(i)}$. This KG encapsulates the entire medical history of the patient, structured in a cohesive and inter-connected manner.

## C.3 Implementation Details of Step 3 (§3.3): BAT GNN

**Attention Initialization.** Keywords we experimented for attention initialization are in Table 4

Table 4: Keyword candidates we attempted for attention initialization. We *highlight* the keywords we finally used in the experiments.

| Task | Conditions | Procedures | Drugs |
|------|-----------|-----------|-------|
| **MT.** | *terminal condition*, *critical diagnosis*, *end-stage*, *life-threatening* | *critical interventions*, *life-saving measures*, *resuscitation*, *emergency procedure* | *palliative medication*, *end-of-life drugs*, *life support drugs*, *emergency meds* |
| **RA.** | *chronic ailment*, *postoperative complication*, *recurrent*, *readmission-prone* | *follow-up procedure*, *secondary intervention*, *post-treatment*, *treatment review* | *maintenance medication*, *postoperative drugs*, *treatment continuation*, *follow-up meds* |
| **LOS** | *acute condition*, *severe diagnosis*, *long-term ailment*, *extended-care diagnosis* | *major surgery*, *intensive procedure*, *long recovery intervention*, *extended hospitalization* | - |
| **Drug.** | *chronic disease*, *acute diagnosis*, *symptomatic*, *treatable condition* | *diagnostic procedure*, *treatment procedure*, *medical intervention*, *drug-indicative procedure* | - |

---

[9]`https://openai.com/blog/new-and-improved-embedding-model`

**Pateint Representations Study.** We detail this in Appendix E.1. Based on the results, we use joint representation in experiments.

**Hyperparameter Study.** We detail this in Appendix G.2.

## C.4 EXPERIMENT ENVIRONMENTS

**Hardware.** All experiments are conducted on a machine equipped with two AMD EPYC 7513 32-Core Processors, 528GB RAM, eight NVIDIA RTX A6000 GPUs, and CUDA 11.7.

**Software.** We implement GRAPHCARE using Python 3.8.13, PyTorch 1.12.0 (Paszke et al., 2019), and PyTorch Geometric 2.3.0 (Fey & Lenssen, 2019). We employ PyHealth (Yang et al., 2023a) to pre-process the EHR data (illustrated in Appendix B). We utilize medical code mappings from ICD-(9/10) to CCS for conditions and procedures, from NDC to ATC (level-3) for drugs. The mapping files are provided by AHRQ (Elixhauser A, 2016) and BioPortal (Noy et al., 2009). We use Gephi (Bastian et al., 2009) for knowledge graph visualization.

## C.5 TRAINING DETAILS

**General Setting.** We split the dataset by 8:1:1 for training/validation/testing data, and we use Adam (Kingma & Ba, 2014) as the optimizer. Based on our hyperparameter study in Appendix G.2, we set learning rate 1e-5, weight decay 1e-5, batch size 4, and hidden dimension 128. All models are trained via 50 epochs over all patient samples, and the early stopping strategy monitored by AUROC with 10 epochs is applied.

**Features for Different Tasks.** We take conditions and procedures as the features for the length-of-stay prediction and drug recommendation and additionally take drugs as features for mortality prediction and readmission prediction.

**Baseline Models.** We use PyHealth (Yang et al., 2023a) pipeline to load the implemented models with their best reported settings. For GRAPHCARE w/ GPS (Rampášek et al. (2022)), we apply LapPE (Kreuzer et al. (2021)) as the Laplacian positional encoding, GINE (Hu et al. (2019)) as the local message-passing mechanism, and Transformer (Vaswani et al. (2017)) for the global attention.

# D KNOWLEDGE GRAPH CONSTRUCTION

In this section, we illustrate our solution to construct a biomedical knowledge graph (KG) for each medical concept by prompting from a large language model (LLM) and sampling a subgraph from a well-established KG.

## D.1 PROMPTING KG FROM LARGE LANGUAGE MODEL

**GPT-KG.** Figure 6 showcases a carefully designed prompt for the retrieval of a biomedical KG from a generative LLM. The main goal of this approach is to leverage the extensive knowledge embedded in the LLM to extract meaningful triples consisting of two entities and a relationship.

In our strategy, we begin with a prompt related to a medical condition, a procedure, or a drug. The LLM is then tasked with generating a list of updates that extrapolate as many relationships as possible from this prompt. Each update is a triple following the format `[ENTITY 1, RELATIONSHIP, ENTITY 2]` where `ENTITY 1` and `ENTITY 2` should be nouns. Our goal is to generate these triples in both breadth (a wide variety of entities and relationships related to the initial term) and depth (following chains of relationships to discover new entities and relationships). The process continues until we obtain a list with 100 KG triples. This prompting-based approach provides a structured, interconnected knowledge graph from the unstructured knowledge embedded in the LLM, which proves especially beneficial for personalized KG generation.

In the experiment, we iterate through the vocabulary of conditions, procedures, and drugs contained in CCS and ATC (level-3) with their code-name mappings[10,11]: $M : e^{\triangle} \leftarrow e$ where $e^{\triangle}$ is the

---

[10]https://www.hcup-us.ahrq.gov/toolssoftware/ccs
[11]https://bioportal.bioontology.org/ontologies/ATC

```python
def ehr_kg_prompting(term, category):

    if category == "condition":
    example = \
    """
    Example:
    prompt: systemic lupus erythematosus
    updates: [[systemic lupus erythematosus, is an, autoimmune condition], [systemic
    lupus erythematosus, may cause, nephritis], [anti-nuclear antigen, is a test for,
    systemic lupus erythematosus], [systemic lupus erythematosus, is treated with,
    steroids], [methylprednisolone, is a, steroid]]
    """
    elif category == "procedure":
    example = \
    """
    Example:
    prompt: endoscopy
    updates: [[endoscopy, is a, medical procedure], [endoscopy, used for, diagnosis],
    [endoscopic biopsy, is a type of, endoscopy], [endoscopic biopsy, can detect,
    ulcers]]
    """
    elif category == "drug":
    example = \
    """
    Example:
    prompt: iobenzamic acid
    updates: [[iobenzamic acid, is a, drug], [iobenzamic acid, may have, side effects],
    [side effects, can include, nausea], [iobenzamic acid, used as, X-ray contrast
    agent], [iobenzamic acid, formula, C16H13I3N2O3]]
    """

    gpt = GPT()
    response = gpt.chat(
    f"""
    Given a prompt (a medical condition/procedure/drug), extrapolate as many
    relationships as possible of it and provide a list of updates.
    The relationships should be helpful for healthcare prediction (e.g., drug
    recommendation, mortality prediction, readmission prediction …)
    Each update should be exactly in format of [ENTITY 1, RELATIONSHIP, ENTITY 2]. The
    relationship is directed, so the order matters.
    Both ENTITY 1 and ENTITY 2 should be noun.
    Any element in [ENTITY 1, RELATIONSHIP, ENTITY 2] should be conclusive, make it as
    short as possible.
    Do this in both breadth and depth. Expand [ENTITY 1, RELATIONSHIP, ENTITY 2] until
    the size reaches 100.

    {example}

    prompt: {term}
    updates:

    """
    )

    # Process the response to triples
    triples = parse(response)

    return triples
```

Figure 6: **Prompting knowledge graphs for medical concepts (EHR terms) from GPT**.

corresponding name for the medical code $e$. For each term $e^{\triangle}$, we input it with its category (either "condition", "procedure" or "drug") to the function `ehr_kg_prompting()` shown in Figure 6 $\chi$ times and compose the graphs of all runs into one, i.e., $G^e = (G_1^e \cup G_2^e \cup ... \cup G_{\chi}^e)$, to obtain more comprehensive graphs.

Table 5: **Expert Evaluation of Knowledge Graph Triples for Medical Concepts.** We assess the quality of triples for randomly selected 50 medical concepts from each coding system (vocabulary). Four metrics: breadth, depth, faithfulness, and bias are used for the evaluation, each scored on a 1-5 scale. *A higher score indicates better performance.* **Breadth** signifies the variety of triples in which the target medical concept features as an entity. **Depth** represents the degree of interconnectedness of triples (e.g., given triple $t_1 : (e_1, r_1, e_2)$ and $t_2 : (e_2, r_2, e_3)$, $t_2$ is an extension of $t_1$). **Faithfulness** quantifies the overall factual accuracy of the triples. We present both the average score and standard deviation for each metric in our evaluation.

| Concept type | Conditions | Procedures | Drugs | Concept type | Conditions | Procedures | Drugs |
|---|---|---|---|---|---|---|---|
| Vocabulary | CCSCM | CCSPROC | ATC-3 | Vocabulary | CCSCM | CCSPROC | ATC-3 |
| Breadth | $4.2 \pm 0.4$ | $3.8 \pm 0.3$ | $4.0 \pm 0.2$ | Breadth | $4.6 \pm 0.2$ | $4.5 \pm 0.3$ | $4.6 \pm 0.2$ |
| Depth | $4.0 \pm 0.3$ | $3.9 \pm 0.2$ | $3.3 \pm 0.4$ | Depth | $3.8 \pm 0.3$ | $3.9 \pm 0.5$ | $4.1 \pm 0.4$ |
| Faithfulness | $4.5 \pm 0.3$ | $4.7 \pm 0.3$ | $4.5 \pm 0.2$ | Faithfulness | $4.8 \pm 0.1$ | $4.9 \pm 0.1$ | $4.6 \pm 0.1$ |

|(a) Evaluation of GPT-KG.| |(b) Evaluation of GPT-UMLS-KG.|

We engaged a medical professional collaborating with us to evaluate the KG triples produced by LLM. The outcomes of this evaluation are presented in Table 5a. As evidenced by the results, the triples generated by GPT-4 exhibit high quality in terms of their breadth, depth, and faithfulness.

Furthermore, after clustering of nodes / edges with $\delta = 0.15$, we futher eliminated 27 out of the 4,626 nodes (clusters) due to their inclusion of inaccurate or potential misleading content, with the help from medical professionals. This measure resulted in the removal of 3,393 KG triples, addressing potential ethical concerns. We also asked medical professionals for their help to remove triples that contain inaccurate, biased, or misleading information, which resulted in the removal of 4,539 triples. This triple filtering process addresses the potential echical concerns.

As a result, we obtained 65,993 non-redundant KG triples with 48,914 unique entities and 8,067 unique relations when we set $\chi = 3$, as shown in Table 6. For future work, we will explore to use this prompting-based method to construct more task-specific KGs, aiming at providing more relevant triples especially beneficial to a certain prediction.

## D.2 SAMPLING SUBGRAPH FROM EXISTING KG

**UMLS-KG.** To extract subgraphs for medical concepts from existing well-established biomedical KG like UMLS (Bodenreider, 2004), we take the following steps:

1. We use `text-embedding-ada-002` to retrieve the word embedding of all entities in the UMLS KG and all concepts contained in the target medical coding system (CCS-CM, CCS-PROC in our case) for conditions and procedures. For drugs (ATC-3), we use the existing ATC-to-UMLS_CUI mapping provided by BioPortal[12].

2. For each medical concept, we search the entity in UMLS that with the most similar word embeeding, and create a mapping from CCS/ATC concept names to those entities.

3. For each UMLS entity in this mapping, we apply subgraph sampling described in the following Algorithm 1.

In Algorithm 1. We have four arguments - medical concept $e$ (in CCS/ATC), source KG $\mathcal{G}$ (UMLS), hop limit $\kappa$ and window size $\epsilon$. In brief, we search all the triples containing $e$ for the first hop and search $\epsilon$ triples containing the other entity for each previous-hop triple. When setting $\kappa = 2$ and $\epsilon = 5$, we obtain 265,587 non-redundant KG triples with 137,845 unique entities and 94 unique relations, as shown in Table 6.

---

[12]https://bioportal.bioontology.org/ontologies

---

**Algorithm 1** Subgraph Sampling

---

1: **procedure** SUBGRAPHSAMPLING(medical concept $e$, KG $\mathcal{G}$, hop limit $\kappa$, window size $\epsilon$)
2:     Initialize an empty list $Q$ and an empty graph $G^e_{\mathrm{sub}(\kappa)} = (\mathcal{V}^e_{\mathrm{sub}(\kappa)}, \mathcal{E}^e_{\mathrm{sub}(\kappa)})$
3:     Add $e$ to $Q$
4:     $\mathcal{V}^e_{\mathrm{sub}(\kappa)} \leftarrow \mathcal{V}^e_{\mathrm{sub}(\kappa)} \cup \{e\}$
5:     **for** $i = 1$ to $\kappa$ **do**
6:         Initialize an empty list $Q_{\mathrm{next}}$
7:         **for all** ent $\in Q$ **do**
8:             **if** $i = 1$ **then**
9:                 Retrieve all triples $(\mathrm{ent}, \mathrm{rel}, \mathrm{ent}')$ or $(\mathrm{ent}', \mathrm{rel}, \mathrm{ent})$ from $\mathcal{G}$
10:            **else**
11:                 Randomly retrieve $\epsilon$ triples $(\mathrm{ent}, \mathrm{rel}, \mathrm{ent}')$ or $(\mathrm{ent}', \mathrm{rel}, \mathrm{ent})$ from $\mathcal{G}$
12:            **end if**
13:            Add retrieved triples to $\mathcal{E}^e_{\mathrm{sub}(\kappa)}$
14:            $\mathcal{V}^e_{\mathrm{sub}(\kappa)} \leftarrow \mathcal{V}^e_{\mathrm{sub}(\kappa)} \cup \{\mathrm{ent}'\}$
15:            Add ent$'$ to $Q_{\mathrm{next}}$
16:         **end for**
17:         $Q \leftarrow Q_{\mathrm{next}}$
18:     **end for**
19:     **return** $G^e_{\mathrm{sub}(\kappa)}$
20: **end procedure**

---

Table 6: **Statistics of GPT-KG (generated through prompting §D.1) and UMLS-KG (extracted through subgraph sampling §D.2)**. We report the data in the format of (# unique nodes, # unique edges, # triples).

| KG | Hyperparameter | Condition | Procedure | Drug | Total |
|---|---|---|---|---|---|
| GPT-KG | $\chi$=3 | (17780, 3633, 22421) | (9636, 1991, 10429) | (26922, 4362, 33380) | (48914, 8067, 65993) |
| UMLS-KG | $\kappa$=1 | (11895, 40, 17747) | (3614, 41, 4158) | (6509, 50, 7547) | (20466, 66, 29334) |
| UMLS-KG | $\kappa$=2, $\epsilon$=5 | (86143, 70, 151294) | (68129, 71, 98817) | (63274, 79, 87267) | (137845, 94, 265587) |

**GPT-UMLS-KG.** By integrating concept-specific KGs produced by GPT-4 with those from UMLS, we constructed the GPT-UMLS-KG. The expert assessment of this amalgamated KG is presented in Table 5b. Notably, compared to GPT-KG, there is an enhancement in quality across all dimensions, consistent with our observations in Figure 3.

## D.3 KNOWLEDGE GRAPHS AFTER CLUSTERING

Table 7: **Statistics of GPT-KG, UMLS-KG, and GPT-UMLS-KG after node/edge clustering**.

| KG | Hyperparameter | # Nodes | # Edges | # Triples |
|---|---|---|---|---|
| GPT-KG | $\chi$=3 | 4599 | 752 | 31325 |
| UMLS-KG | $\kappa$=1 | 3053 | 40 | 12421 |
| UMLS-KG | $\kappa$=2, $\epsilon$=5 | 10805 | 54 | 81073 |
| GPT-UMLS-KG | $\chi$=3, $\kappa$=1 | 6355 | 774 | 40496 |
| GPT-UMLS-KG | $\chi$=3, $\kappa$=2, $\epsilon$=5 | 12284 | 785 | 104460 |

In Table 7, we present the KGs following the node/edge clustering process detailed in §3.1, with a set value of $\delta = 0.15$ (as optimized in Appendix G.1). We note a notably low triple union between GPT-KG and UMLS-KG. This suggests that the knowledge from one can significantly complement the other. Consequently, GPT-UMLS-KG is poised to outperform either of the two individual KGs. This inference is empirically supported by our results displayed in Figure 3.

## D.4 ANALYSIS ON GPT-UMLS-KG

Table 8 illustrates the impact of the two GPT-UMLS-KG variants on enhancing the performance of EHR predictions. It is evident that the performance significantly improves when $\kappa = 1$, while

Table 8: Comparison of the performance gain from the GPT-UMLS-KG with 1-hop and 2-hop concept-specific subgraph sampled from UMLS.

| | MIMIC-III | | | | MIMIC-IV | | | |
|---|---|---|---|---|---|---|---|---|
| KG | MT. | RA. | LOS | Drug. | MT. | RA. | LOS | Drug. |
| GPT-UMLS-KG ($\chi$=3, $\kappa$=1) | **70.3** | **69.7** | **37.5** | **66.8** | **73.1** | **68.5** | **34.2** | **63.9** |
| GPT-UMLS-KG ($\chi$=3, $\kappa$=2, $\epsilon$=5) | 68.4 | 67.2 | 35.4 | 63.4 | 72.6 | 66.7 | 33.4 | 62.2 |

the performance with $\kappa = 2$ sometimes even fails to surpass the baseline performance without any external knowledge (e.g. outperformed by RETAIN on MIMIC-III drug recommendation task), as we compare the performance to Table 2. Possible explanations for this outcome:

- The constrained window size ($\epsilon$) increases the randomness of the triples sampled after the initial 1-hop, resulting in a proliferation of isolated nodes and the formation of isolated clusters. This situation poses a considerable challenge for the GRAPHCARE model to effectively learn from the knowledge graph.

- The increased randomness is very likely to exclude critical triples originating from a source node, leading to the propagation of irrelevant knowledge triples (noise) that ultimately detrimentally affect the model's performance.

Therefore, developing a more effective method to sample more useful triples from existing KGs becomes one of our future works.

## E  "PATIENT AS A GRAPH" AND "PATIENT AS A NODE"

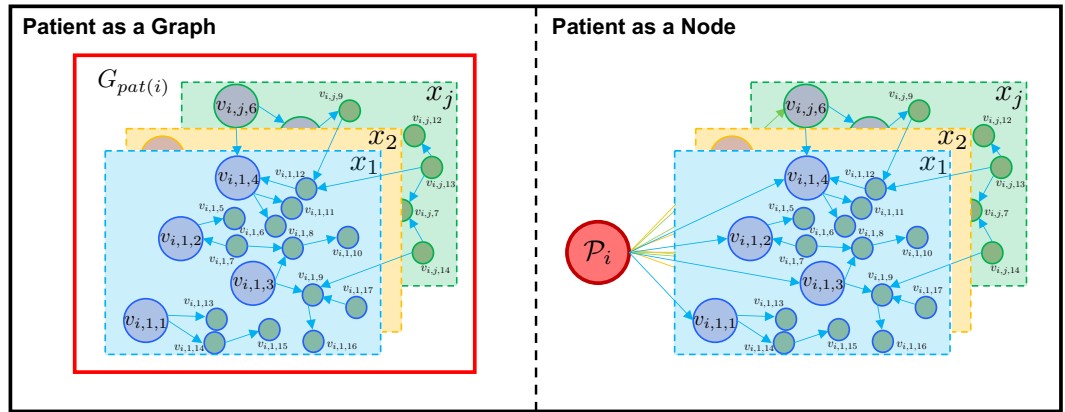

Figure 7: **Comparison of two patient representations in GRAPHCARE.** *Left*: patient as a graph covering the information in all nodes. *Right*: patient as a node only connected to the nodes of the direct medical codes (the larger ones) recorded in the EHR dataset. $x_j$ denotes the $j$-th for patient $i$. $v_{i,j,k}$ denotes the $k$-th node of the $j$-th visit for patient $i$. The connections among nodes are either inner-visit or across-visit.

Figure 7 presents two different patient representations. When viewed as a graph, the patient representation aims to encapsulate a comprehensive summary of all nodes, thus providing a broad overview of information. However, this approach may also include more noise due to its extensive scope. In contrast, when a patient is represented as a node, the information is aggregated solely from directly corresponding EHR nodes. This approach ensures a precise match with the patient's EHR data, offering a more accurate, albeit narrower, representation. Although this method provides a more focused insight, it also inevitably discards information from other nodes, thus potentially losing broader contextual data. Therefore, the choice between these two representations hinges on the balance between precision and the extent of information required. In our experiment, we introduce a joint embedding composed by concatenating those two embeddings, as a balanced solution.

Figure 8: **Performance of healthcare predictions with three types of patient representations (§3.3)**: (1) **graph** - patient graph embedding obtained through mean pooling of node embedding; (2) **node** - patient node embedding connected to the direct EHR node; (3) **joint** - embedding concatenated by (1) and (2). We use GPT-KG to perform this analysis.

### E.1 PATIENT REPRESENTATION LEARNING.

We further discuss the performance of different patient representations in GRAPHCARE, as depicted in Figure 8. We calculate the average over 20 independent runs for each type of patient representation and for each task. Our observations reveal that the patient node embedding presents more stability as it is computed by averaging the direct EHR nodes. These nodes are rich in precise information, thereby reducing noise, but they offer limited global information across the graph. On the other hand, patient graph embedding consistently exhibits the most significant variance, with the largest distance observed between the maximum and minimum outliers. Despite capturing a broader scope of information, the graph embedding performs less effectively due to the increased noise. This is attributed to its derivation method that averages all node embeddings within a patient's personalized KG, inherently incorporating a more diverse and complex set of information. The joint embedding operates as a balanced compromise between the node and graph embeddings. It allows GRAPHCARE to learn from both local and global information. Despite the increased noise, the joint embedding provides an enriched context that improves the model's understanding and prediction capabilities.

## F  IMPORTANCE SCORE

To provide insights into the GRAPHCARE's decision-making process, we propose an interpretation method that computes the importance scores for the entities and relationships in the personalized knowledge graph. We first compute the entity importance scores as the sum of the product of node-level attention weights $\alpha_{i,j,k}$ and visit-level attention weights $\beta_{i,j}$ (obtained by Eq (3)) over all visits, and relationship importance scores as the edge weights $\mathrm{w}^{(l)}_{\mathcal{R}\langle k,k'\rangle}$ summed over all GNN layers:

$$\mathrm{I}^{\mathrm{ent}}_{i,k} = \sum_{l=1}^{L-1} \sum_{j=1}^{J} \beta^{(l)}_{i,j} \alpha^{(l)}_{i,j,k}, \quad \mathrm{I}^{\mathrm{rel}}_{i,k,k'} = \sum_{l=1}^{L-1} \mathrm{w}^{(l)}_{\mathcal{R}\langle k,k'\rangle}, \tag{6}$$

where $\mathrm{I}^{\mathrm{ent}}_{i,k}$ is the importance score of entity $k$ and $\mathrm{I}^{\mathrm{rel}}_{i,k,k'}$ is the importance score of the relationship between entities $k$ and $k'$. To identify the most crucial entities and relationships, we can also compute the top $K$ entities and relationships with the highest importance scores, denoted as $s$ in descending order of their importance:

$$\mathcal{T}^{\mathrm{ent}}_{i,K} = \{s \mid s \in \mathrm{I}^{\mathrm{ent}}_i, s \geq \mathrm{I}^{\mathrm{ent}}_{i,(K)}\}, \quad \mathcal{T}^{\mathrm{rel}}_{i,K} = \{s \mid s \in \mathrm{I}^{\mathrm{rel}}_i, s \geq \mathrm{I}^{\mathrm{rel}}_{i,(K)}\}, \tag{7}$$

where $\mathrm{I}^{\mathrm{ent}}_{i,(K)}$ and $\mathrm{I}^{\mathrm{rel}}_{i,(K)}$ are the $K$-th highest importance scores for entities and relationships, respectively, $\mathcal{T}^{\mathrm{ent}}_{i,K}$ and $\mathcal{T}^{\mathrm{rel}}_{i,K}$ represent the top $K$ entities and relationships for patient $i$, respectively. By analyzing the top entities and relationships, we can gain a better understanding of the model's decision-making process and identify the most influential factors in its predictions.

## G  HYPER-PARAMETER TUNING

Given that our GRAPHCARE utilizes personalized knowledge graphs (KGs) as inputs for healthcare predictions, the representativeness of the constructed graphs becomes critical in the prediction process. The quality and structure of these KGs can significantly influence the performance of

our predictive model, underlining the importance of thoroughly investigating the hyperparameters involved in their construction and subsequent analysis via the BAT GNN model. Therefore, we meticulously examine both the hyperparameters for KG node/edge clustering and those for our proposed BAT Graph Neural Network (GNN) model. We use GPT-KG as the external knowledge and use validation set of EHR data for a more efficient parameter searching.

## G.1 HYPER-PARAMETERS FOR CLUSTERING

Table 9: Clustering hyperparameter tuning. Tested on GPT-KG.

| Threshold $\delta$ | # Cluster | Mortality | | Readmission | |
| --- | --- | --- | --- | --- | --- |
| | | AUPRC | AUROC | AUPRC | AUROC |
| 0.05 | 29681 | 12.2 | 61.3 | 65.5 | 63.5 |
| 0.1 | 14662 | 13.3 | 65.2 | 70.0 | 67.4 |
| 0.15 | 4599 | **15.7** | **69.6** | **72.6** | **68.9** |
| 0.2 | 883 | 13.9 | 67.8 | 67.8 | 66.7 |

| Threshold $\delta$ | # Cluster | LOS | | Drug Rec. | |
| --- | --- | --- | --- | --- | --- |
| | | F1-score | AUROC | F1-score | AUROC |
| 0.05 | 15094 | 32.8 | 77.4 | 62.1 | 94.2 |
| 0.1 | 7941 | 34.7 | 79.7 | 64.8 | 94.5 |
| 0.15 | 2755 | **36.6** | **80.2** | **65.2** | **95.1** |
| 0.2 | 589 | 34.1 | 77.9 | 63.8 | 94.3 |

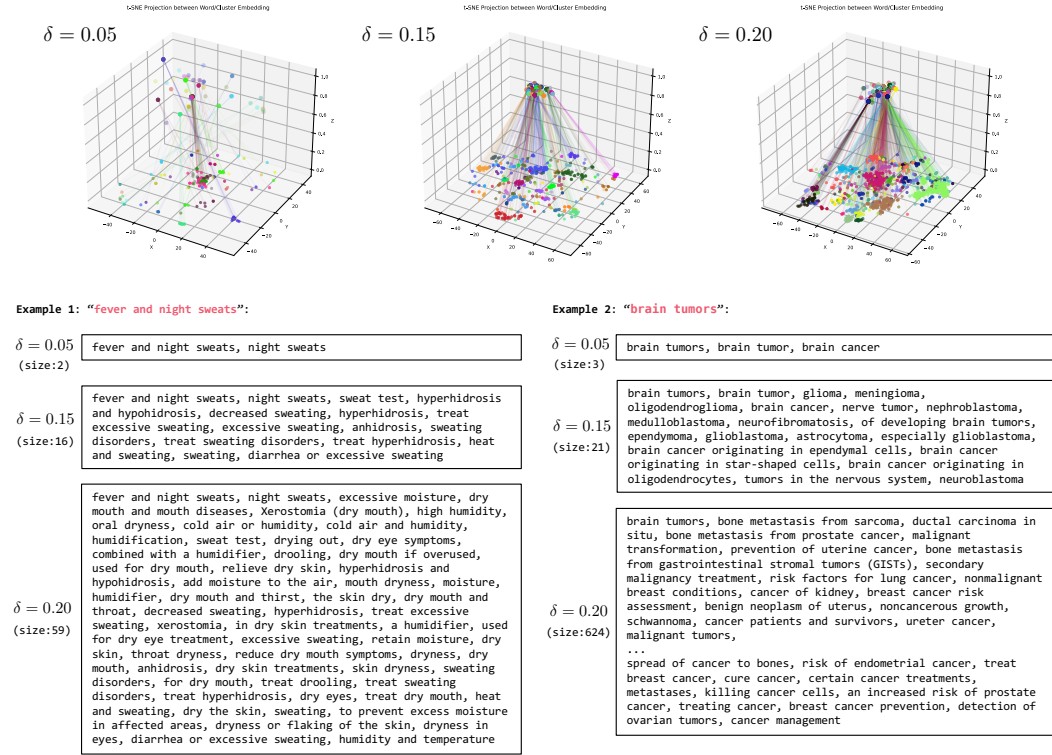

Figure 9: **Comparison between the node clustering over GPT-KG and UMLS-KG.** *Above*: we random sample 80 clusters with each distance threshold $\delta$ applied. Each figure visually represents the clustering of words, with color consistency denoting membership to the same cluster. *Below*: we provide two examples of the clusters with different $\delta$'s for the given words ("fever and night sweats" and "brain tumors").

Table 9 presents the performance of GRAPHCARE across four tasks on the MIMIC-III dataset, with varying agglomerative clustering distance thresholds $\delta \in \{0.05, 0.1, 0.15, 0.2\}$. We evaluate the

performance with the GPT-KG. The results reveal that the model achieves optimal performance when $\delta = 0.15$. This outcome can be attributed to the following reasons: when $\delta$ is small, nodes of high similarity may be incorrectly classified as distinct, complicating the learning process for the model. Conversely, if $\delta$ is large, dissimilar nodes could be inaccurately clustered together, which further challenges the training process. Examples in Figure 9 further demonstrate our findings.

The examples presented in Figure 9 illustrate the significant influence of the distance threshold $\delta$ on the semantic coherence of the clusters. When $\delta = 0.20$, the clusters tend to incorporate words that aren't strongly semantically related to the given word. For instance, "humidity" is inappropriately grouped with "fever and night sweats", and "breast cancer" is incorrectly associated with "brain tumors". Conversely, when $\delta = 0.05$, the restrictive threshold fails to capture several words closely related to the given word, such as the absence of "heat and sweating" in the cluster for "fever and night sweats", and "glioma" for "brain tumors".

Striking an optimal balance, when $\delta = 0.15$, the resulting clusters exhibit a desirable semantic coherence. Most words within these clusters are meaningfully related to the given word. This observation underlines the importance of selecting an appropriate $\delta$ value to ensure the extraction of semantically consistent and comprehensive clusters. This is a pivotal step, as the quality of these clusters has a direct impact on subsequent healthcare prediction tasks, which rely on the KG constructed through this process.

## G.2    Hyper-parameters for the Bi-attention GNN

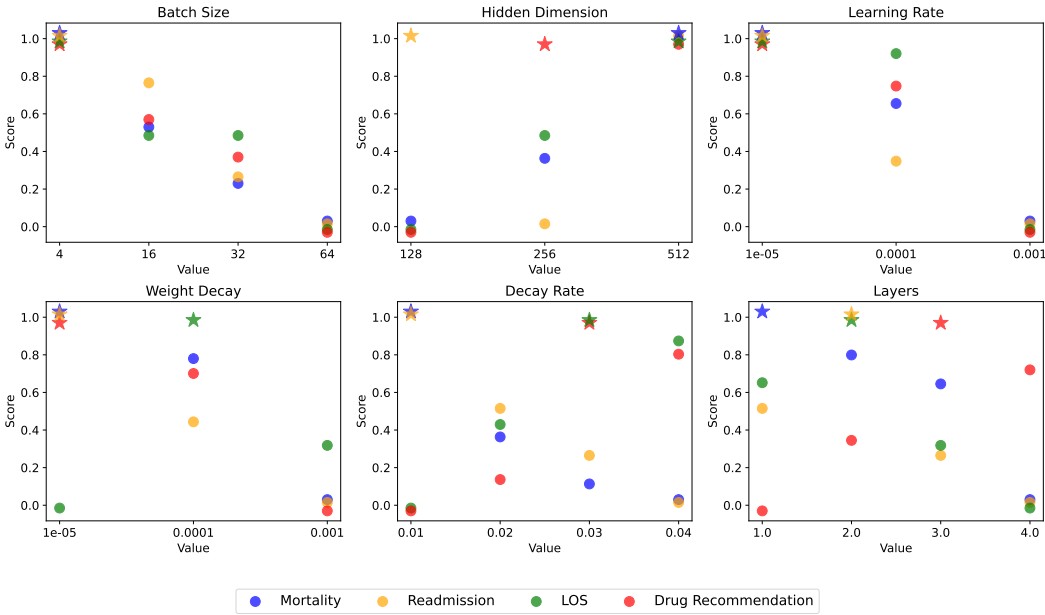

Figure 10: **Hyper-parameter tuning.** We tune each parameter while keeping other hyperparameters fixed as their default values (batch size: 4, hidden dimension: 128, learning rate: 1e-5, weight decay: 1e-5, decay rate: 0.01, layers: 1). *Score* denotes the normalized AUROC in the range of [0, 1], which shows the relative performance of a specific setting compared to others. The best value of each hyperparameter for each task is labeled as a star.

| Task | Batch Size | Hidden Dimension | Learning Rate | Weight Decay | Decay Rate | Layers |
|------|-----------|------------------|---------------|--------------|-----------|--------|
| Mortality | 4 | 128 | 1e-5 | 1e-5 | 0.01 | 1 |
| Readmission | 4 | 128 | 1e-5 | 1e-5 | 0.01 | 2 |
| Length-Of-Stay (LOS) | 4 | 128 | 1e-5 | 1e-5 | 0.03 | 2 |
| Drug Recommendation | 4 | 128 | 1e-5 | 1e-5 | 0.03 | 3 |

Table 10: Hyper-parameters for the BAT GNN model for different tasks.

For our proposed BAT GNN we tune the following hyper-parameters: batch size in {4, 16, 32, 64}, hidden dimension in {128, 256, 512}, learning rate in {1e-3, 1e-4, 1e-5}, weight decay in {1e-3,

1e-4, 1e-5}, decay rate $\gamma$ in {0.01, 0.02, 0.03, 0.04} and number of layers $L$ in {1, 2, 3, 4}. We show the tuning detail in Figure 10. The hyper-parameters employed throughout the experiments presented in this paper are consolidated and presented in Table 10. For the sake of maintaining a more fair and balanced comparison, we align the batch size, hidden dimension, learning rate, and weight decay with those of the baseline models.

## H  NOTATION TABLE

For clarity, we have attached a notation table here, describing all symbols used in the main paper.

Table 11: Notations and Descriptions in GRAPHCARE

| Notation | Description |
|---|---|
| **Notations in Step 1 (§3.1)** | |
| $e \in \{\mathbf{c}, \mathbf{p}, \mathbf{d}\}$ | A medical concept in {conditions, procedures, drugs} |
| $|\mathbf{c}|, |\mathbf{p}|, |\mathbf{d}|$ | Sizes of sets of medical concepts |
| $G^e, \mathcal{V}^e, \mathcal{E}^e$ | KG, nodes, and edges for each medical concept |
| $G^e_{\text{LLM}(\chi)}$ | KG for each medical concept, obtained through prompting LLM $\chi$ times |
| $\mathcal{V}^e_{\text{LLM}(\chi)}, \mathcal{E}^e_{\text{LLM}(\chi)}$ | Nodes and edges in the KG obtained through prompting LLM $\chi$ times (Appendix D.1) |
| $G^e_{\text{sub}(\kappa)}$ | $\kappa$-hop subgraph for a concept, obtained though subgraph sampling (Appendix D.2) |
| $\mathcal{V}^e_{\text{sub}(\kappa)}, \mathcal{E}^e_{\text{sub}(\kappa)}$ | Nodes and edges in the $\kappa$-hop subgraph obtained though subgraph sampling |
| $\mathcal{C}_\mathcal{V}, \mathcal{C}_\mathcal{E}$ | Clustering mappings for nodes and edges |
| $\delta$ | Distance threshold of cosine similarity |
| $G, \mathcal{V}, \mathcal{E}$ | Global graph composed by all concept-specific KGs, and its nodes and edges |
| $G', \mathcal{V}', \mathcal{E}'$ | New global graph, nodes and edges after clustering |
| $w$ | Dimension of the word embedding |
| $\mathbf{H}^\mathcal{V}, \mathbf{H}^\mathcal{R}$ | (Initial) node and edge embeddings |
| **Notations in Step 2 (§3.2)** | |
| $\mathcal{P}$ | Patient node |
| $G_{\text{pat}}, \mathcal{V}_{\text{pat}}, \mathcal{E}_{\text{pat}}$ | Personalized KG for a patient |
| $\epsilon$ | Edge connecting patient node and the node of medical concepts in patient's EHR |
| $G_{i,j}$ | Visit-subgraph for the $j$-th visit of the patient $i$ |
| $v_{i,j,k}$ | The $k$-th node in the $j$-th visit of the patient $i$ |
| $(v_{i,j,k} \leftrightarrow v_{i,j',k'})$ | The edge between the node $v_{i,j,k}$ and $v_{i,j',k'}$ |
| $\mathcal{E}_{\text{inter}}$ | Interconnected edges across visit-subgraphs |
| **Notations in Step 3 (§3.3)** | |
| $\mathbf{h}_k^{(l+1)}$ | Updated node representation of node $k$ at $(l+1)$-th layer of GNN |
| $\sigma$ | Activation function |
| $\mathbf{W}^{(l)}$ | Learnable weight matrix at $l$-th layer |
| $\text{AGGREGATE}^{(l)}$ | Function to aggregate node representations |
| $\mathbf{b}^{(l)}$ | Bias vector at $l$-th layer |
| $\mathcal{N}(k)$ | Neighbors of node $k$ |
| $\mathbf{h}_{i,j,k}$ | Hidden embedding of $k$-th node in $j$-th visit-subgraph of patient |
| $\mathbf{h}_{(i,j,k)\leftrightarrow(i,j',k')}$ | Hidden embedding of edge between nodes $v_{i,j,k}$ and $v_{i,j'k'}$ |
| $\mathbf{W}_v, \mathbf{W}_r$ | Learnable matrices in $\mathbb{R}^{w \times q}$ |
| $\mathbf{b}_v, \mathbf{b}_r$ | Learnable vectors in $\mathbb{R}^q$ |
| $\mathbf{h}^\mathcal{V}_{(i,j,k)}, \mathbf{h}^\mathcal{R}_{(i,j,k)\leftrightarrow(i,j',k')}$ | Input embeddings of the node and edge |
| $q$ | Size of the hidden embedding |
| $\alpha_{i,j,k}, \beta_{i,j}$ | Node-level and visit-level attention weights |
| $\mathbf{g}_{i,j}, \mathbf{G}_i$ | Multi-hot vector and matrix for visit-subgraph and patient's graph |
| $\mathbf{W}_\alpha, \mathbf{w}_\beta$ | Learnable parameters for node-level attention and visit-level attention |
| $\mathbf{b}_\alpha, \mathbf{b}_\beta$ | Bias vectors for node-level attention and visit-level attention |
| $\boldsymbol{\lambda}$ | Decay coefficient vector |
| $\gamma$ | Decay rate |
| $\mathbf{w}_{\text{tf}}$ | Word embedding of a keyword for task-feature pair |
| $w_m$ | Computed weight for $m$-th node in global graph $G'$ |
| $\mathbf{h}^{(L)}_{i,j,k}$ | Node embeddings of final layer for predictions |
| $\mathbf{h}^{G_{\text{pat}}}_i$ | Patient graph embedding |
| $\mathbf{h}^\mathcal{P}_i$ | Patient node embedding |
| $\mathbb{1}^\Delta_{i,j,k}$ | Binary label indicating direct medical concept |
| $\mathbf{z}^{\text{graph}}_i, \mathbf{z}^{\text{node}}_i, \mathbf{z}^{\text{joint}}_i$ | Logits from different embeddings after MLP |

