# OpenReview forum: "GraphCare: Enhancing Healthcare Predictions with Personalized Knowledge Graphs"
_ICLR.cc/2024/Conference — ICLR 2024 poster_

### Official Review · Reviewer_qgPW · 2023-10-29

**Soundness:** 3 good
**Presentation:** 3 good
**Contribution:** 3 good
**Rating:** 6
**Confidence:** 4

**Summary:**

This paper focuses on clinical event prediction such as mortality and readmission prediction and drug recommendation using EHR datasets. The authors propose to build knowledge graphs for medical concepts in EHR datasets. They first extract relations for every medical concept and then apply node/edge clustering based on word embeddings. Finally, they design a node-level and visit-level attention method to predict medical events. Experimental results show that the proposed method can largely enhance the event prediction performance.

**Strengths:**

1) This paper is well-written and easy to follow.

2) The proposed knowledge graph for medical concept built by LLM is interesting and novel.

3) This paper conducts extensive experiments including ablation study and interpreta0bility analysis.

**Weaknesses:**

1) The discussion on related work could be more comprehensive. For instance, there are several new developments on the below aspects:

* Recent work about knowledge graph extraction using LLMs.

[1] Yao, Liang, Chengsheng Mao, and Yuan Luo. "KG-BERT: BERT for knowledge graph completion." arXiv preprint arXiv:1909.03193 (2019).

[2] Wang, Chenguang, Xiao Liu, and Dawn Song. "Language models are open knowledge graphs." arXiv preprint arXiv:2010.11967 (2020).

[3] Chen, Chen, et al. "Knowledge Is Flat: A Seq2Seq Generative Framework for Various Knowledge Graph Completion." Proceedings of the 29th International Conference on Computational Linguistics. 2022.

[4] Lovelace, Justin, and Carolyn Rose. "A Framework for Adapting Pre-Trained Language Models to Knowledge Graph Completion." Proceedings of the 2022 Conference on Empirical Methods in Natural Language Processing. 2022.

[5] Chen, Chen, et al. "Dipping PLMs Sauce: Bridging Structure and Text for Effective Knowledge Graph Completion via Conditional Soft Prompting." Findings of the Association for Computational Linguistics: ACL 2023. 2023.

* Regarding the bi-attention method, the below paper may be related:

[6] Lu, Chang, Chandan K. Reddy, and Yue Ning. "Self-supervised graph learning with hyperbolic embedding for temporal health event prediction." IEEE Transactions on Cybernetics (2021).

* The latest baseline, StageNet, in the evaluation tasks is published in 2020. New papers have been published recently:

[6] Yang, Kai, et al. "KerPrint: local-global knowledge graph enhanced diagnosis prediction for retrospective and prospective interpretations." Proceedings of the AAAI Conference on Artificial Intelligence. Vol. 37. No. 4. 2023.

[7] Lu, Chang, et al. "Context-aware health event prediction via transition functions on dynamic disease graphs." Proceedings of the AAAI Conference on Artificial Intelligence. Vol. 36. No. 4. 2022.

[8] Ma, Xinyu, et al. "Patient Health Representation Learning via Correlational Sparse Prior of Medical Features." IEEE Transactions on Knowledge and Data Engineering (2022).

[9] Xu, Yongxin, et al. "SeqCare: Sequential Training with External Medical Knowledge Graph for Diagnosis Prediction in Healthcare Data." Proceedings of the ACM Web Conference 2023. 2023.

[10] Sun, Ziyou, et al. "EHR2HG: Modeling of EHRs Data Based on Hypergraphs for Disease Prediction." 2022 IEEE International Conference on Bioinformatics and Biomedicine (BIBM). IEEE, 2022.

Although these baselines may not be used in all tasks 1, 2, and 3, the base model can be interchangeable. The authors should pick at least a few of them for comparison.

2) The knowledge graph building is somewhat unclear. A knowledge graph usually contains edges with multiple types. However, the edge type is not reflected in KG extraction, edge clustering, and the graph neural network.

3) One suggestion is to compare more clustering methods.

**Questions:**

1) In node/edge clustering, a node can be a phrase. What is the word embedding of such nodes?

2) Above Equation (2), the authors said that reducing the size of node/edge embedding can handle the sparsity problem. What is the sparsity problem here, and why can the node/edge embedding have such a sparsity problem, given that they are dense vectors? If there indeed is a sparsity problem, can we decrease the initial embedding size?

---

> ### Author Response · Authors · 2023-11-13
> **Author Response to Reviewer qgPW - Part 1**
>
> Thank you for recognizing the clarity and composition of our paper, the novelty of our LLM-built knowledge graph for medical concepts, and the comprehensiveness of our experimental work and interpretability analysis. We address your concerns and answer your questions below. We also uploaded a revision and used blue to mark the new changes.
>
> ****
>
> **W1: The discussion on related work could be more comprehensive.**
>
> **(1) For recent works about KG extraction using LLM:** We have taken your advice and added the related papers to our related work section, which are highlighted in $\textcolor{blue}{blue}$.
>
> **(2) For the paper related to the bi-attention method [1]:**
>
> Thanks for letting us know about this paper! We have reviewed the work by Lu et al. [1] and acknowledge its relevance to our bi-attention mechanism. We have accordingly updated our related work section to include this reference. While there are conceptual similarities between their multi-level attention mechanism and our bi-attention mechanism, several key distinctions set our methodology apart:
>
> 1. **Simplicity and Efficiency**: Our model diverges from Lu et al.’s approach by calculating node-level and visit-level attention scores through a multi-hot matrix representation. In this schema, each row corresponds to a distinct multi-hot vector that signifies the presence of nodes within a particular visit, with subsequent rows differentiating among various visits. This design streamlines the process, as it utilizes sparser representations that inherently require fewer dense matrix operations. Consequently, our method enhances practical computational efficiency and facilitates parallel processing.
> 2. **Independence**: Another advantage of our approach is the independent computation of node-level and visit-level attention. By decoupling these two, we prevent the potential propagation of errors from one level to another, enhancing the robustness of the model.
>
> We have conducted an experiment to showcase the performance difference between these two attention computation methods applied to GraphCare on MIMIC-III mortality prediction:
>
> |                                            | AUPRC | AUROC | Time (1 epoch) |
> | ------------------------------------------ | ----- | ----- | -------------- |
> | **GraphCare (ours)**                       | 16.7  | 70.3  | 78 secs        |
> | **Graphcare w/ [1]'s attention mechanism** | 16.4  | 69.6  | 127 secs       |
>
> The comparison clearly demonstrates that our bi-attention mechanism not only significantly enhances inference time efficiency but also achieves slightly higher prediction accuracy.
>
> **(3) Adding new baselines:**
>
> Thanks for introducing these new papers! We will include some of them in comparison. ~~Due to the intricate process involved in accurately reproducing these methodologies, we anticipate needing a few days (before the author response deadline) to integrate them into our testing framework thoroughly~~ We have shown the new results in ***"Update to Part 1 (New Baselines)"*** below.
>
> ****
>
> **W2: The knowledge graph building is somewhat unclear. A knowledge graph usually contains edges with multiple types. However, the edge type is not reflected in KG extraction, edge clustering, and the graph neural network.**
>
> Thank you for your observation regarding the construction of our knowledge graph (KG). We included our KG construction details in ***Appendix D***. Our approach to KG assembly through LLM prompting leads to a variable number of edge types, which may differ from traditional KGs that typically feature a predefined edge taxonomy. The statistics regarding the number of edge types after clustering are revealed in ***Table 7 in Appendix D.3***.
>
> ****
>
> **W3: One suggestion is to compare more clustering methods.**
>
> Thank you for the valuable suggestion to explore a broader range of clustering methods. We agree that a comprehensive comparison, including hyperparameter tuning, is crucial for a thorough evaluation of clustering strategies. Due to the extensive nature of these experiments, we will diligently work on this analysis and aim to present the results in our final revision.
>
> **(References are attached in Part 2)**

---

> > ### Author Response · Authors · 2023-11-13
> > **Author Response to Reviewer qgPW - Part 2**
> >
> > **Q1: In node/edge clustering, a node can be a phrase. What is the word embedding of such nodes?**
> >
> > As described by Appendix C.1, we retrieve embeddings using the $\texttt{text-embedding-ada-002}$ provided by the OpenAI service [2], which generates a single vector embedding for input text regardless of token count. Consequently, even when a node or an edge represents a phrase, the resultant embedding is a singular vector with a dimensionality of 1536. This dimension is consistent with that of embeddings for single-word nodes/edges.
> >
> > **Q2: Above Equation (2), the authors said that reducing the size of node/edge embedding can handle the sparsity problem. What is the sparsity problem here, and why can the node/edge embedding have such a sparsity problem, given that they are dense vectors? If there indeed is a sparsity problem, can we decrease the initial embedding size?**
> >
> > We apologize for any confusion caused by the mention of a 'sparsity problem.' This term was included by mistake and has been corrected in our revised manuscript. The reduction in node/edge embedding size is intended to enhance the model's efficiency. We do not encounter a sparsity issue as the embeddings employed are dense. Regarding the initial size of the embeddings, we use the dimensions provided by the LLM, which are fixed at 1536.
> >
> >
> >
> >
> > **References:**
> >
> > [1] Lu, Chang, Chandan K. Reddy, and Yue Ning. "Self-supervised graph learning with hyperbolic embedding for temporal health event prediction." IEEE Transactions on Cybernetics (2021).
> >
> > [2] OpenAI. (n.d.). Embeddings Guide. Link: https://platform.openai.com/docs/guides/embeddings/. Last visit: Nov. 13, 2023.

---

> > > ### Author Response · Authors · 2023-11-19
> > > **Author Response to Reviewer qgPW - Update to Part 1 (New Baselines)**
> > >
> > > We have tested the baselines [3, 4, 5, 6] you recommended on the MIMIC-III dataset (100 runs). We did not test [7] as we could not find a publicly released code of it. The results are shown as follows.
> > >
> > > **Mortality Prediction:**
> > >
> > > |              | AUPRC      | AUROC      |
> > > | ------------ | ---------- | ---------- |
> > > | KeyPrint [3] | 13.5 (0.7) | 64.7 (0.6) |
> > > | Chet [4]     | 12.7 (0.2) | 61.9 (0.5) |
> > > | SAFARI [5]   | 12.9 (0.9) | 62.2 (0.6) |
> > > | SeqCare [6]  | 14.0 (0.5) | 66.5 (0.8) |
> > > | GraphCare    | 16.7 (0.5) | 70.3 (0.5) |
> > >
> > > **Readmission Prediction:**
> > >
> > > |              | AUPRC      | AUROC      |
> > > | ------------ | ---------- | ---------- |
> > > | KeyPrint [3] | 70.2 (0.7) | 67.2 (0.6) |
> > > | Chet [4]     | 68.8 (0.8) | 66.1 (0.5) |
> > > | SAFARI [5]   | 68.4 (0.6) | 66.0 (0.3) |
> > > | SeqCare [6]  | 70.7 (0.4) | 67.0 (0.6) |
> > > | GraphCare    | 73.4 (0.4) | 69.7 (0.5) |
> > >
> > > **Length of Stay Prediction:**
> > >
> > > |              | AUROC      | Kappa      | Accuracy   | F1-Score   |
> > > | ------------ | ---------- | ---------- | ---------- | ---------- |
> > > | KeyPrint [3] | 79.5 (0.4) | 28.3 (0.5) | 41.6 (0.4) | 35.9 (0.5) |
> > > | Chet [4]     | 78.5 (0.2) | 26.7 (0.3) | 41.0 (0.3) | 35.3 (0.4) |
> > > | SAFARI [5]   | 78.3 (0.2) | 26.4 (0.4) | 40.6 (0.4) | 35.2 (0.3) |
> > > | SeqCare [6]  | 78.9 (0.2) | 27.8 (0.3) | 41.5 (0.3) | 35.7 (0.3) |
> > > | GraphCare    | 81.4 (0.3) | 29.5 (0.4) | 43.2 (0.4) | 37.5 (0.2) |
> > >
> > > **Drug Recommendation:**
> > >
> > > |              | AUPRC      | AUROC      | F1-Score   | Jaccard    |
> > > | ------------ | ---------- | ---------- | ---------- | ---------- |
> > > | KeyPrint [3] | 78.9 (0.3) | 95.0 (0.1) | 65.9 (0.3) | 49.2 (0.4) |
> > > | Chet [4]     | 77.7 (0.1) | 94.6 (0.1) | 63.6 (0.3) | 48.5 (0.3) |
> > > | SAFARI [5]   | 78.2 (0.2) | 94.7 (0.1) | 65.2 (0.4) | 49.0 (0.4) |
> > > | SeqCare [6]  | 79.1 (0.2) | 95.1 (0.1) | 66.0 (0.2) | 49.4 (0.3) |
> > > | GraphCare    | 80.2 (0.2) | 95.5 (0.1) | 66.8 (0.2) | 49.7 (0.3) |
> > >
> > >
> > > We will further test them on MIMIC-IV and include the results in Table 2 in the final revision of our paper.
> > >
> > > ****
> > >
> > > **References**
> > >
> > > [3] Yang, Kai, et al. "KerPrint: local-global knowledge graph enhanced diagnosis prediction for retrospective and prospective interpretations." Proceedings of the AAAI Conference on Artificial Intelligence. Vol. 37. No. 4. 2023.
> > >
> > > [4] Lu, Chang, et al. "Context-aware health event prediction via transition functions on dynamic diseasegraphs." Proceedings of the AAAI Conference on Artificial Intelligence. Vol. 36. No. 4. 2022.
> > >
> > > [5] Ma, Xinyu, et al. "Patient Health Representation Learning via Correlational Sparse Prior of Medical Features." IEEE Transactions on Knowledge and Data Engineering (2022).
> > >
> > > [6] Xu, Yongxin, et al. "SeqCare: Sequential Training with External Medical Knowledge Graph for Diagnosis Prediction in Healthcare Data." Proceedings of the ACM Web Conference 2023. 2023.
> > >
> > > [7] Sun, Ziyou, et al. "EHR2HG: Modeling of EHRs Data Based on Hypergraphs for Disease Prediction." 2022 IEEEInternational Conference on Bioinformatics and Biomedicine (BIBM). IEEE, 2022.

---

> > > > ### Comment · Reviewer_qgPW · 2023-11-21
> > > >
> > > > Thank you for the response and the new experiments. I think my concerns are addressed.

---

> > > > > ### Author Response · Authors · 2023-11-22
> > > > > **Thank you!**
> > > > >
> > > > > We are delighted to know that your concerns have been resolved! In our final revision, we will incorporate the latest results to enrich our analysis. We also greatly appreciate your valuable suggestions regarding the inclusion of related works on knowledge extraction from language models, as well as incorporating more up-to-date baseline comparisons. These additions will undoubtedly contribute to making our paper more thorough and comprehensive.

---

### Official Review · Reviewer_baD4 · 2023-11-01

**Soundness:** 3 good
**Presentation:** 2 fair
**Contribution:** 3 good
**Rating:** 6
**Confidence:** 4

**Summary:**

The paper proposes a framework that leverages external medical knowledge generated by LLM to create patient-specific knowledge graphs (KGs) for improving healthcare predictions. In experiments on MIMIC-III and MIMIC-IV datasets, GraphCare outperforms baselines in multiple tasks, including mortality, readmission, length of stay, and drug recommendation. These results highlight the potential of external KGs in healthcare predictions, particularly in scenarios with limited data.

**Strengths:**

- There are extensive experiments including different prediction tasks, ablation studies on different parts of the model, KG sizes, and data sizes. These experiments are helpful to understand the impact of the proposed method.

- The paper introduces a novel approach, leveraging Language Models (LM) to create a Knowledge Graph (KG). This innovative idea simplifies the KG generation process and ensures that the extracted knowledge is semantically rich and interpretable.

- The knowledge graph provide interpretability of the connections among different EHR nodes.

**Weaknesses:**

- The performance of mortality prediction is questionable. While previous literature has reported AUPRC values exceeding 20 on the MIMIC dataset, the results presented in Table 2 fall notably short of this benchmark [1]. What leads to such a gap?

- The paper delves into various engineering details, such as Knowledge Graph (KG) construction and attention initialization, without offering sufficient clarity. The notation in sec 3.3 are too complicated to follow. Many of them are randomly presented without definition.

- The paper only compares with the baseline methods without leveraging external information / knowledge. However, it will be more important to compare with knowledge graph generated by other methods [2].


[1] Purushotham, S., Meng, C., Che, Z., & Liu, Y. (2017). Benchmark of Deep Learning Models on Large Healthcare MIMIC Datasets.
[2] Rotmensch, M., Halpern, Y., Tlimat, A. et al. Learning a Health Knowledge Graph from Electronic Medical Records. Sci Rep 7, 5994 (2017).

**Questions:**

- What is κ-hop subgraph in Section 1?

- How is distance threshold δ selected for node and edge clustering?

- What is AGGREGATE in equation 1?

---

> ### Author Response · Authors · 2023-11-13
> **Author Response to Reviewer baD4 - Part 1**
>
> Thank you for recognizing the breadth of our experiments, the innovative use of LLM in KG generation, and the interpretability provided by our knowledge graph in GraphCare. We address your concerns and answer your questions below. We also uploaded a revision and used blue to mark the new changes.
>
> ****
>
> **W1: The performance of mortality prediction is questionable. While previous literature has reported AUPRC values exceeding 20 on the MIMIC dataset, the results presented in Table 2 fall notably short of this benchmark [1]. What leads to such a gap?**
>
> Thank you for highlighting the performance discrepancy in mortality prediction as compared to previous literature. The observed gap can be attributed to several key differences:
>
> 1. **Task Definition:** Our study defines mortality prediction as forecasting the outcome of the next visit based on previous visits, aligning with studies [3, 4]. This contrasts with [1], which considers mortality within specific time windows and employs a different feature selection approach.
> 2. **Input Features:** Unlike [1], which selects features from multiple tables (*chartevents, labevents, outputevents, icustays, diagnoses_icd*) in the MIMIC-III datasets, our study uses features from the (*diagnoses_icd, procedures_icd, prescriptions*) tables, consistent with the setting in [3].
> 3. **Medical Coding:** We use the CCS and ATC (level-3) coding systems, as mapped from ICD-9 and NDC respectively, a well-recognized setting in literature [3, 6, 7]. This differs from the ICD-9 coding system used in [1].
>
> These factors collectively contribute to the differences in performance results.
>
> ****
>
> **W2: The paper delves into various engineering details, such as Knowledge Graph (KG) construction and attention initialization, without offering sufficient clarity. The notation in sec 3.3 is too complicated to follow. Many of them are randomly presented without definition.**
>
> Thank you for pointing out the need for greater clarity in our explanation of the Knowledge Graph (KG) construction and attention initialization, as well as the complexity of notations in Section 3.3.
>
> **Engineering Details:** To enhance clarity, we have comprehensively updated ***Appendix C***. This appendix now contains detailed explanations of each step mentioned in Section 3, ensuring a clearer understanding of the engineering aspects of our work.
>
> **Notations Definition:** We understand that the notations may have seemed complex. To address this, we have included a comprehensive notation-description table in ***Appendix H - Table 11*** in the updated version of our paper. This table is designed to provide clear definitions and explanations for the notations used, aiding in easier comprehension of the paper.
>
> We hope these updates make the technical aspects of our paper more accessible and easier to follow.
>
> **W3: The paper only compares with the baseline methods without leveraging external information/knowledge. However, it will be more important to compare with the knowledge graph generated by other methods [2].**
>
> Thank you for suggesting a comparison with the method described in [2], which utilizes the Google Health Knowledge Graph (GHKG). We acknowledge the value of such a comparison; however, there are practical constraints to consider:
>
> 1. **Availability of GHKG:** The GHKG, a key component of the method in [2], is not publicly available. Access to this resource is crucial for replicating or comparing with their approach. Unfortunately, obtaining permission from Google for access to the GHKG is not feasible within the limited time frame of this discussion window.
>
> 2. **Comparison with Other Graph-Based Methods:** While we were unable to compare with the GHKG-based method, we have conducted comparisons with other graph-based methods, GRAM [8] and GAMENet [9]. These comparisons are detailed in Table 2 and Figure 2 of our paper, providing insights into the performance of our approach relative to other graph-based methods that do not rely on proprietary datasets like the GHKG.
>
> We believe these comparisons, although not with the specific method you suggested, still offer valuable insights into the efficacy of our approach in the context of available graph-based methods in the field.
>
> **(References are attached in Part 2)**

---

> > ### Author Response · Authors · 2023-11-13
> > **Author Response to Reviewer baD4 - Part 2**
> >
> > **Q1: What is κ-hop subgraph in Section 1?**
> >
> > The $\kappa$-hop subgraph concept refers to a graph sampling technique where we start with a specific entity, represented by the concept $e$, and then include all connected entities within $\kappa$ steps (or hops) from $e$. This process effectively captures the local neighborhood around the concept $e$ up to a distance of $\kappa$ edges.
> >
> > For example, in a 2-hop subgraph, we would include not only the entities directly connected to $e$ but also those that are two connections away. This method allows us to construct a localized view of the knowledge graph, focusing on the immediate and near-immediate relational context of a given entity.
> >
> > The $\kappa$-hop subgraph sampling is further detailed in ***Appendix D.2***.
> >
> >
> > **Q2: How is distance threshold δ selected for node and edge clustering?**
> >
> > We detailed the tuning of threshold $\delta$ in ***Appendix G.1*** where we experimented with thresholds ranging from 0.05 to 0.20 to determine the most effective value. The performance comparison in ***Table 9*** and the examples shown in **Figure 9** both indicate that a threshold of 0.15 yields the best balance.
> >
> >
> > **Q3: What is AGGREGATE in equation 1?**
> >
> > The term "AGGREGATE" in Equation 1 refers to a method used in Graph Neural Networks (GNNs) to combine, or 'aggregate', information from neighboring nodes/edges of a graph. In simple terms, for each node (a point in the graph), we look at its neighbors (other connected points) and gather their information to update the node's own information.
> >
> > In our work, this process of aggregation not only involves bringing together information from neighboring nodes (including the node itself) but also incorporates the type and significance of their connections. This is detailed in Equation 4, where we show how both node information and connection (edge) information are combined to enhance the understanding of each node within the graph.
> >
> >
> >
> > **References**
> >
> > [1] Purushotham, S., Meng, C., Che, Z., & Liu, Y. (2017). Benchmark of Deep Learning Models on Large Healthcare MIMIC Datasets.
> >
> > [2] Rotmensch, M., Halpern, Y., Tlimat, A. et al. Learning a Health Knowledge Graph from Electronic Medical Records. Sci Rep 7, 5994 (2017).
> >
> > [3] Yang, C., Wu, Z., Jiang, P., Lin, Z., Gao, J., Danek, B.P. and Sun, J., PyHealth: A Deep Learning Toolkit for Healthcare Applications. KDD (2023).
> >
> > [4] Choi, E., Xu, Z., Li, Y., Dusenberry, M., Flores, G., Xue, E. and Dai, A., Learning the graphical structure of electronic health records with graph convolutional transformer. AAAI (2020).
> >
> > [5] A. for Healthcare Research and Quality. Clinical classifications software (ccs) for icd-9-cm. link: https: //www.hcup-us.ahrq.gov/toolssoftware/ccs/ccs.jsp.
> >
> > [6] Choi, E., Bahadori, M.T., Sun, J., Kulas, J., Schuetz, A. and Stewart, W., Retain: An interpretable predictive model for healthcare using reverse time attention mechanism. NeurIPS (2016).
> >
> > [7] Choi, E., Schuetz, A., Stewart, W.F. and Sun, J., Using recurrent neural network models for early detection of heart failure onset. JAMIA (2017).
> >
> > [8] Choi, E., Bahadori, M.T., Song, L., Stewart, W.F. and Sun, J., GRAM: graph-based attention model for healthcare representation learning. KDD (2017).
> >
> > [9] Shang, J., Xiao, C., Ma, T., Li, H. and Sun, J., Gamenet: Graph augmented memory networks for recommending medication combination.AAAI (2019).

---

> > > ### Comment · Reviewer_baD4 · 2023-11-22
> > >
> > > Thanks for the detailed explanation! They answered my questions.
> > >
> > > The authors can also provide more technical details in the appendix in the main text in the future revision. It will help the readers a lot.

---

> > > > ### Author Response · Authors · 2023-11-22
> > > > **Thank you for your suggestion!**
> > > >
> > > > We are pleased to hear that your questions have been satisfactorily answered! Your suggestion to include more technical details in the main text is indeed valuable. We will ensure to incorporate these additional specifics in our final revision, aiming to enhance the clarity of the presentation.

---

### Official Review · Reviewer_ZFn4 · 2023-11-02

**Soundness:** 3 good
**Presentation:** 3 good
**Contribution:** 3 good
**Rating:** 6
**Confidence:** 4

**Summary:**

The paper introduces GraphCare, a framework that utilizes external knowledge graphs to enhance healthcare predictions and decision-making. This framework generates personalized knowledge graphs for patients, thus improving accuracy and performance in crucial healthcare prediction tasks. The methodology of GraphCare, including its process of extracting knowledge from large language models and external biomedical KGs, is discussed in detail. Moreover, the paper presents experimental results on the MIMIC-III and MIMIC-IV datasets, demonstrating the framework's effectiveness in enhancing prediction accuracy and explainability. Finally, the limitations and potential future directions of GraphCare are discussed.

**Strengths:**

- GraphCare generates personalized knowledge graphs for patients, enhancing the accuracy of predictions. This insight is compelling, as personalized predictions are often more accurate than general ones.
- The paper presents experimental results on the MIMIC-III and MIMIC-IV datasets, which demonstrate the framework's effectiveness in improving prediction accuracy and explainability. The results indicate that the proposed methods surpass other state-of-the-art approaches.
- The paper acknowledges the limitations and suggests future directions for GraphCare, indicating the authors' awareness of potential challenges and their consideration of future improvements.
- The authors conduct comprehensive ablation studies on the backbone models, adding robustness to their findings.

**Weaknesses:**

- The complexity of the proposed framework presents challenges for implementation. The paper would benefit from a more comprehensive explanation of the implementation specifics of the framework.
- While the authors discuss the interpretability of GraphCare in Section 4.3, the depth of analysis is confined to observations on entity connections and their impact on model performance. It would be beneficial if the framework could uncover deeper insights in line with existing literature, or identify longer-distance relationships that might inspire new scientific discoveries.
- In relation to the chosen experimental tasks, it would be helpful if the authors could elaborate on their selection criteria. Is there a particular preference for the information that the constructed graph helps in elucidating?

**Questions:**

- The authors are encouraged to provide additional results on the quality assurance of their constructed personalized knowledge graphs. The significance of this is underscored by the direct impact these graphs have on the results.
- The authors should also address the concerns highlighted in the 'weaknesses' section of the review.

**Details Of Ethics Concerns:**

- The authors should address potential privacy concerns associated with integrating diverse medical records over time within the constructed patient-level graph.
- It is important for the authors to consider whether the constructed knowledge graph may contain biases or potentially misleading information that could negatively affect the accuracy of predictions.

---

> ### Author Response · Authors · 2023-11-13
> **Author Response to Reviewer ZFn4 - Part 1**
>
> Thank you for recognizing the strengths of GraphCare, especially its effectiveness with the MIMIC datasets and our comprehensive approach to limitations and future directions.  We address your concerns and answer your questions below. We also uploaded a revision and used blue to mark the new changes.
>
> ****
>
> **W1: The complexity of the proposed framework presents challenges for implementation. The paper would benefit from a more comprehensive explanation of the implementation specifics of the framework.**
>
> Thank you for your feedback regarding the complexity of our framework and the need for clearer implementation details. We agree that a more detailed explanation would enhance the understanding of our framework.
>
> 1. In our original submission, we provided a summary of the implementation details in the README.md file within the supplementary material's code folder. This was intended to offer quick access to the necessary information for replicating our framework.
>
> 2. Following your suggestion, we have revised and expanded the ***Appendix C - Implementation Details*** section. This update aims to provide a more thorough and organized explanation of the implementation process, ensuring that readers can easily understand and apply our framework. We have also created a comprehensive notation table in ***Appendix H*** to enhance the clarity of our paper.
>
> We believe these enhancements will greatly assist in comprehending and utilizing our proposed framework effectively.
>
> **W2: While the authors discuss the interpretability of GraphCare in Section 4.3, the depth of analysis is confined to observations on entity connections and their impact on model performance. It would be beneficial if the framework could uncover deeper insights in line with existing literature, or identify longer-distance relationships that might inspire new scientific discoveries.**
>
> Thank you for your valuable feedback regarding the depth of our interpretability study in GraphCare. We agree that exploring deeper insights and longer-distance relationships could significantly enhance its application in real-world scenarios and foster scientific discoveries.
>
> To address your point, GraphCare could indeed be adapted to investigate critical factors in various scenarios. For instance, in identifying key causes of a disease, we can focus on the disease as the main prediction target, systematically examining and pruning long-distance relationships to pinpoint the most crucial ones. Similarly, we could study how these relationships might correlate with a patient's age or other demographic details. The potential applications are practically endless.
>
> It's essential to recognize that conducting such in-depth analysis, especially in a clinical setting, demands comprehensive validation. This might involve integrating sophisticated fact-checking methods with advanced information retrieval techniques. While this level of detailed scrutiny is crucial, it extends beyond the primary scope of our current study.
>
> Our main contribution lies in developing GraphCare as a versatile framework. This framework is instrumental in constructing personalized KGs and adeptly managing diverse prediction tasks. Nonetheless, we acknowledge the significance of your suggestion. It aligns well with our objectives, and we are eager to explore it as a central aspect of our future research.

---

> > ### Author Response · Authors · 2023-11-13
> > **Author Response to Reviewer ZFn4 - Part 2**
> >
> > **W3: Further elaboration on the selection criteria for the experimental tasks is needed to understand the preference for the information elucidated by the constructed graph.**
> >
> > Thank you for highlighting the need for further explanation on our selection of experimental tasks in relation to the constructed graph in GraphCare. Our decision was strategically aligned with showcasing the graph's ability to extract and utilize complex patient data in various healthcare scenarios:
> >
> > 1. ***Mortality Prediction:*** This task was chosen to demonstrate the graph's capability in processing critical, binary outcomes. The graph's structure and embeddings are particularly adept at highlighting key indicators relevant to mortality, which is a crucial, high-stakes decision-making scenario in clinical settings.
> >
> > 2. ***Readmission Prediction:*** This task leverages the graph's ability to capture short-term patient trajectories, utilizing the temporal data embedded in the graph. It showcases the framework's proficiency in interpreting dynamic patient data for predicting readmissions, a key aspect of hospital resource management and patient care.
> >
> > 3. ***Length-Of-Stay Prediction:*** The task of predicting ICU stay lengths is aligned with the graph's ability to categorize and analyze time-sensitive data. It exemplifies the graph's utility in operational healthcare planning, where understanding and predicting patient flow and resource needs are essential.
> >
> > 4. ***Drug Recommendation:*** This complex task was specifically chosen to highlight the graph's strength in multi-dimensional analysis. It demonstrates how the graph can integrate and interpret a multitude of patient-specific data points, essential in the multi-faceted decision process of medication prescription. This task underscores the graph's capacity to handle multiple predictions simultaneously, a scenario that mirrors the complexities of real-world clinical decision-making.
> >
> > Each of these tasks was selected not only for their relevance to healthcare but also for their alignment with the strengths of our graph in processing diverse, intricate, and multi-dimensional patient data. This careful alignment ensures that our experimental tasks comprehensively evaluate and showcase the capabilities of GraphCare in interpreting and utilizing the rich information provided by the constructed graph.
> >
> > ****
> >
> > **Q1: Provide additional results on the quality assurance of constructed personalized knowledge graphs, considering their direct impact on results.** and
> >
> > **Ethics Concerns 2: Consider biases or misleading information in the constructed knowledge graph that could affect prediction accuracy.**
> >
> > Thank you for your question about the quality assurance of our constructed personalized knowledge graphs (KGs) and your concern regarding potential biases or misleading information.
> >
> > 1. **Quality Assurance:** In Figure 3, we have shown the effect of different KG sizes on performance, which indirectly reflects the quality of the KGs in terms of their completeness. We believe completeness is a critical measure of quality. To ensure this, as detailed in Appendix D.1, we employed a meticulous process of human-annotated triple selection, highlighted in blue, to eliminate any inaccurate or potentially biased information from the KGs.
> >
> > 2. **Addressing Biases and Misleading Information:** To further address your concern about biases and misleading information, we conducted additional tests. These involved reintroducing the previously removed 'negative' information (inaccurate, biased, and misleading) into the KGs. The results, tested on the MIMIC-III dataset using GPT-UMLS-KG, are as follows (with the bracketed results representing the cleaned KG):
> >
> > |                            | AUPRC       | AUROC       |
> > | -------------------------- | ----------- | ----------- |
> > | **Mortality Prediction**   | 16.2 (16.7) | 69.7 (70.3) |
> > | **Readmission Prediction** | 73.1 (73.4) | 69.2 (69.7) |
> >
> > |                               | AUROC       | Kappa       | accuracy    | F1-Score    |
> > | ----------------------------- | ----------- | ----------- | ----------- | ----------- |
> > | **Length-of-Stay Prediction** | 80.9 (81.4) | 29.0 (29.5) | 42.9 (43.2) | 37.0 (37.5) |
> >
> > |                         | AUPRC       | AUROC       | F1-Score    | Jaccard     |
> > | ----------------------- | ----------- | ----------- | ----------- | ----------- |
> > | **Drug Recommendation** | 79.5 (80.2) | 95.2 (95.5) | 66.0 (66.8) | 49.1 (49.7) |
> >
> >   These results demonstrate that while the inclusion of negative information in the KG can slightly reduce performance, GraphCare still consistently outperforms all baselines. This indicates the framework's resilience to noisy graphs and its ability to handle some level of imperfect data.
> >
> > We hope these explanations and results adequately address your concerns about KG quality and the steps we've taken to mitigate potential biases and inaccuracies in our framework.

---

> > > ### Author Response · Authors · 2023-11-13
> > > **Author Response to Reviewer ZFn4 - Part 3**
> > >
> > > **Ethics Concerns 1: Address potential privacy concerns associated with integrating diverse medical records over time within the patient-level graph.**
> > >
> > > Thank you for raising the important issue of privacy concerns associated with integrating medical records over time within patient-level graphs in GraphCare. We take privacy considerations seriously and have designed our framework with this in mind, as detailed in Appendix A.1.
> > >
> > > 1. **No Global Patient Graph:** In GraphCare, we avoid building a global graph that links different patient graphs together. This approach inherently safeguards against privacy risks associated with interlinking individual patient data.
> > >
> > > 2. **LLM Use and Patient Data:** The utilization of Language Learning Models (LLMs) in our framework is strictly for building medical concept-specific Knowledge Graphs (KGs). This process does not involve direct interaction with any individual patient's data. We iterate through all concepts in the medical coding system (such as CCS, ICD) to generate their respective KGs using LLMs, and these KGs are stored locally.
> > >
> > > Therefore, the design of GraphCare is such that it prioritizes patient privacy and data security, making sure that individual patient records are not directly accessed or linked globally, eliminating potential privacy concerns.

---

> > > > ### Comment · Reviewer_ZFn4 · 2023-11-22
> > > >
> > > > Thanks for providing the additional experiments and explanations. I think they solve my concerns.

---

> > > > > ### Author Response · Authors · 2023-11-22
> > > > > **Thank you!**
> > > > >
> > > > > We are pleased to hear that your concerns have been addressed satisfactorily! We will integrate new results into our final revision. Additionally, we will delve deeper into the interpretability analysis of GraphCare, as per your insightful suggestion.

---

### Meta-Review · Area_Chair_uxLh · 2023-12-06

**Metareview:**

a)   The paper introduces GraphCare, a framework that utilizes external knowledge graphs to enhance healthcare predictions and decision-making.  Empirical studies are done.
b)   Empirical studies with ablation studies.  Interesting approach.  Future work discussed.
c)   More related work needed, and some issues with comparison with prior results.   Some other questions addressed.

**Justification For Why Not Higher Score:**

no one was in support of the paper, all reviewers we marginal

**Justification For Why Not Lower Score:**

All reviews were in favour of accept

---

### Decision · Program_Chairs · 2024-01-16

Accept (poster)